# New Materials Used for the Development of Anion-Selective Electrodes—A Review

**DOI:** 10.3390/ma16175779

**Published:** 2023-08-23

**Authors:** Cecylia Wardak, Klaudia Morawska, Karolina Pietrzak

**Affiliations:** 1Department of Analytical Chemistry, Institute of Chemical Sciences, Faculty of Chemistry, Maria Curie-Sklodowska University, Maria Curie-Sklodowska Sq. 3, 20-031 Lublin, Poland; klaudiamorawska0905@gmail.com; 2Department of Food and Nutrition, Medical University of Lublin, 4a Chodzki Str., 20-093 Lublin, Poland; karolina.pietrzak@umlub.pl

**Keywords:** ion-selective electrode, potentiometry, anion ionophore, solid contact, carbon paste ion-selective electrode

## Abstract

Ion-selective electrodes are a popular analytical tool useful in the analysis of cations and anions in environmental, industrial and clinical samples. This paper presents an overview of new materials used for the preparation of anion-sensitive ion-selective electrodes during the last five years. Design variants of anion-sensitive electrodes, their advantages and disadvantages as well as research methods used to assess their parameters and analytical usefulness are presented. The work is divided into chapters according to the type of ion to which the electrode is selective. Characteristics of new ionophores used as the electroactive component of ion-sensitive membranes and other materials used to achieve improvement of sensor performance (e.g., nanomaterials, composite and hybrid materials) are presented. Analytical parameters of the electrodes presented in the paper are collected in tables, which allows for easy comparison of different variants of electrodes sensitive to the same ion.

## 1. Introduction

Ion-selective electrodes (ISEs) are sensors used in potentiometric measurements. They are also called membrane electrodes due to the presence of an ion-selective membrane, which is one of the most important elements of each ISE. Because of the presence of the ionophore in the membrane, the electrode is sensitive to changes in the activity of particular ions in solutions [1]. The development of ion-selective sensors began more than 100 years ago when at the beginning of the 20th century, Cramer invented a glass electrode, which, for quite a long time (until the 1930s), was successfully used for analytical measurements. Haber and Klemensiewicz were also working on ISEs at the beginning of the last century [2]. Extremely important for potentiometry was the year 1966, when Frant and Ross announced their pioneering discovery of the fluoride ion-selective electrode [3]. Ion-selective sensors were improved in subsequent years by researchers such as Bloch, Moody, Thomas, Bakker and Sokalski. However, the breakthrough in the field of potentiometry came in 1971 when Freiser and Cattrall invented the first solid contact ion-selective electrode (SC-ISE), which they called a coated wire electrode (CWE). It was the first ion-selective electrode in which there was no internal solution. This completely revolutionized the field of potentiometry and opened the way to a number of new construction possibilities [4]. A brief history of ISEs is presented in Figure 1. 

Their discovery contributed to the development of SC-ISEs, which, compared to classical ion-selective electrodes, have a number of advantages undoubtedly due to the elimination of the internal solution. As a result of this action, the design of these electrodes has been simplified and downsized, and, consequently, the production costs of these measuring instruments have been significantly reduced [5,6].

By comparing the differences in the design of classic ion-selective electrodes with a liquid contact and ion-selective electrodes with a solid contact (Figure 2), it can be observed that SC-ISEs have an ion-selective membrane and an inner electrode just like classical ISEs. SC-ISEs differ from classical electrodes because of the absence of an internal electrolyte and the presence of solid contact in the form of an intermediate layer between the internal electrode and the ion-selective membrane. The solid contact that is present in SC-ISEs provides the transport of ions and the conversion of their signal into an electrical signal [7,8]. It fulfills the functions of the internal solution in classical electrodes, the presence of which forces the measurement in a vertical position and is the cause of difficulties in the miniaturization of these electrodes.

The complete elimination of the internal electrolyte solution in the SC-ISEs enables not only the miniaturization of these sensors but also often obtains a lower limit of detection (LOD) even to nanomole concentrations [9,10]. This is a consequence of the fact that in the absence of an internal electrolyte solution, there is no flow of ions from the internal solution (containing usually a high concentration of these ions) through the membrane into the sample solution. In addition, the lack of an internal solution eliminates the operations related to it: the problem of its leakage, the presence of air in the electrode and the need to work in a vertical position. SC-ISEs show greater mechanical resistance. This is due to the fact that the membrane of the electrodes with solid contact is usually thicker and placed on a hard surface. This makes them easier to store and transport, which is especially important for field measurements. [11,12]. The electrodes are easy to obtain in small sizes and in any shape. This allows the construction of multi-sensor measurement platforms, e.g., electronic tongues [13] or wearable sensors in the form of wristbands or elements of clothing [4]. The most common disadvantage of SC-ISEs is insufficient potential stability related to inadequate mutual adhesion of the used materials [14] and the formation of an unfavorable water layer between the membrane and the inner electrode [15]. 

The research in the area of ion-selective electrodes is still being intensively conducted and has resulted in the development of sensors with better analytical performance and/or simpler operations. These include two main research directions: the development of new active substances to obtain more selective ion-sensitive membranes and/or new primarily electroconductive materials, which are used in the construction of electrodes without an internal electrolyte solution to improve the process of charge transfer between the membrane and the internal electron conductor. Considering the type of ion to which the electrode is sensitive, there is definitely less work involving anionic electrodes. First and foremost, this is associated with the fact that, for anions, the number of commercially available ionophores is much smaller. This fact is also an inspiration to search for new compounds that can fulfill this function in the membrane.

In the literature on ion-selective electrodes, there are almost no review articles focusing on anion-selective electrodes. Only one short review has been published in the last 20 years in which the authors focused on ionophores used in anion-sensitive membranes [16].

This review presents recent developments in the field of ion-selective electrodes sensitive to inorganic anions. It describes 102 anionic ion-selective electrodes used in potentiometric measurements developed over the last five years. Among them were 67 electrodes with solid contact and 35 with liquid contact. ISEs sensitive to 18 different ions were described, which include anions such as NO_3_^−^, F^−^, Cl^−^, Br^−^, I^−^, ClO_4_^−^, S^2−^, SO_3_^2−^, SO_4_^2−^, H_2_PO_4_^−^, HPO_4_^2−^, PO_4_^3−^, SCN^−^, AsO_4_^3−^, BO_3_^3−^, CH_3_COO^−^, CO_3_^2−^ and SiO_3_^2−^. This paper is a comprehensive overview containing the characteristics of new materials used in the construction of ISEs, including both materials used as active substances of the ion-selective membrane, as well as materials used to improve the efficiency and/or easier and more universal use of ISEs. 

## 2. Constructional Variants of Anionic Ion-Selective Electrodes

The most important component of any ion-selective electrode is the ion-sensitive membrane. The most important component of the membrane from the point of view of electrode operation is the ionophore, which gives it sensitivity and selectivity to a specific ion. The material used as an ionophore should selectively bind a given ion in the membrane environment. In the case of anion-sensitive electrodes, three types of membranes can be distinguished: polymeric membranes based on PVC, crystalline membranes composed of insoluble salts of the determined ion and composite membranes containing the active ingredient combined in various ways with other materials. Polymeric and crystalline membranes are still used in classic liquid-contact electrodes; however, a much larger group includes a variant of electrodes without internal electrolyte solution called all-solid-state ion-selective electrodes (ASS-ISEs). In recent years, they have become, among ISEs, one of the most studied groups of analytical measuring instruments. Depending on their construction, ASS-ISEs can be classified as follows:-CWE, CDE—a coated wire/coated disc electrode, which was originally a platinum wire coated with a layer of PVC membrane. CWEs/CDEs are created by applying, for example, a polymer membrane or crystal membrane on a wire/disc used as the inner electrode. Such sensors are now rarely used. In publications, they are studied as a comparative system to assess the effectiveness of solid contact or modification of the membrane composition.-SC-ISE—solid contact ISE is a type of electrode in which an intermediate layer of material, generally called solid contact, is introduced between the ion-selective membrane and the inner electrode. Conductive polymers, carbon nanomaterials, metal and metal oxide nanoparticles and composite materials are most often used as intermediate layers.-SP-ISE—a single-piece ion-selective electrode with a membrane in which the modifying material is dispersed or dissolved. An ion-selective electrode is created by depositing a membrane cocktail of conductive polymers or other conductive materials, e.g., carbon nanotubes, onto a surface that is usually a glassy carbon electrode [17,18]. Carbon paste ion-selective electrodes can also be included in this group of electrodes in which the active component of the membrane is mixed with graphite powder and binder material, and the mixture is packed in the sensor holder with an internal electrode placed inside (often a copper wire). In order to improve their work, the composition of the paste electrode membrane is also modified by the addition of various materials, which are often carbon nanomaterials. A comparison of the construction of different ASS-ISEs electrodes is shown in Figure 3.

All-solid-state constructions, except CWEs and CDEs, provide a reduction of potential drift and improve stability as well as reproducibility of the potential. The additional solid contact materials and conductive materials used for membrane modification increase the electrical capacitance and improve the ion-electron conductivity. To achieve these goals, a system with solid contact must fulfill three important conditions, which have been defined by Nikolski and Materov: (1) The current passing through the sample during the measurement should be significantly lower than the exchange current, which should be characterized by high values. (2) The chemical indifference towards the other interferents present in the sample—there must be no side reactions taking place alongside the main electrode reaction. (3) The equilibrium of the ion-electron conductivity should be reversible and stable [19].

Two of the most disadvantageous properties of CWEs and CDEs are their weak reproducibility and stability of the potential. Besides this, the formation of an aqueous layer between the ion-selective membrane and the internal (“lead”) electrode is a fairly common problem, which in turn results in the generation of a higher potential drift and a shorter electrode lifetime. This problem can be significantly minimalized by the above-mentioned modifications. In this case, the hydrophobic materials prove to be very suitable.

## 3. Research Methods Used to Assess New ISEs

Newly developed ion-selective electrodes are subjected to numerous tests to determine their parameters and assess their analytical usefulness. The basic measurements carried out in each case include the production of a calibration curve from which the LOD, measuring range and slope of the characteristic are determined. Equally important are the selectivity tests consisting of determining the values of selectivity coefficients towards interfering ions. These measurements are absolutely necessary when a new active substance is introduced into the electrode membrane. The most commonly used methods for determining selectivity coefficients are those recommended by IUPAC, i.e., the separate solution method (SSM) and the fixed interference method (FIM) [20].

In order to check the effectiveness of the introduced electroconductive materials in the form of an intermediate layer or as a membrane component, the potential drift tests are performed: the potential change under zero current conditions and the estimated E^0^ potential stability in time [21]. Chronopotentiometry and electrochemical impedance spectroscopy measurements also provide valuable information in this area as they make it possible to determine the electrical capacitance of the electrodes and their resistance, which significantly influence the stability of the electrode potential [15,18,22]. Water layer tests are also performed according to the procedure proposed by the Pretsch group [15].

## 4. Ion-Selective Electrodes Sensitive to Nitrates Ions

The monitoring of nitrate ions is very important, which is why new tools are being developed to make their identification more accurate and simple. Determining the concentration of nitrate ions is crucial because an excess of nitrates in the human diet is harmful to health and can cause, for example, problems with the cardiovascular system and dysfunction of the intestines and the rest of the gastrointestinal system. It can also lead to pathological conditions such as cancer. We have collected 19 nitrate ion-selective electrodes that were developed over the past five years. Almost all of the nitrate electrodes were solid contact electrodes, with the exception of two that had liquid contact.

For the determination of nitrates, a few groups of scientists proposed electrodes with a solid contact, which had TDMAN (tridodecylmethylammonium nitrate) as the ionophore. They were different from each other, for example, in the inner electrode and material of the intermediate layer. In article [23], the structure of the electrode was developed by using screen-printing technology, in which the working electrode was a graphite electrode deposited from a graphite-based ink, was described. Graphite was used as the transducer layer due to its stability, as well as its good conductive and hydrophobic properties. The slope of the characteristic of −55.4 mV/decade slightly deviated from the Nernst value, and the obtained linearity range for the tested microelectrode was not very wide (2.9 × 10^−4^–1.7 × 10^−1^ M). The limit of detection was quite high, so measurements of low concentrations could be problematic. The advantage of this electrode is its good repeatability. 

The second electrode with TDMAN as the ionophore was described in the publication [24], where the ISE based on hydrophobic laser-induced graphene (LIG) as an intermediate layer was presented. The hydrophobic LIG was created using a polyimide substrate and a double lasing process. A study using the XPS (X-ray photoelectron spectroscopy) technique was carried out to check the composition of the LIG, and the static contact water angle of 135.5 ± 0.7° was determined. The slope of the characteristic equaled −58.17 mV/decade and was much closer to the Nernst response in comparison to the electrode described as the first. The ranges of the linearity were comparable, but the detection limit extended to micromole values, which was a significantly better result. This electrode was successfully applied in the monitoring of surface water quality. 

The authors of another work [25] proposed a novel electrode in which a new nanocomposite consisting of poly(3-octylthiophene) and molybdenum disulfide (POT-MoS_2_) was used as the intermediate layer. A gold electrode (AuE) was used as the basic electrode. The SC-ISE exhibited an over-Nerstian response, while the other parameters were comparable to the ISEs described above. The parameters of the nanocomposite-modified electrodes were not remarkably different from those for ISEs based on MoS_2_ or POT. Nevertheless, the combination of POT-MoS_2_ gave the best results, i.e., the slope, LOD or potential stability. An additional advantage of the newly proposed composite is its high hydrophobicity and redox properties. By replacing the membrane, this electrode can be used for the determination of other ions, i.e., potassium, phosphate or sulfate ions.

Subsequent electrodes having the same type of ionophore—TDMAN—were presented in the work [26,27]. In the publication [26], the solid contact ion-selective electrodes with different types of transducer layers (TLs) were described. The TL was constituted by polyaniline nanofibers doped with Cl^−^ (PANINFs-Cl^−^) or NO_3_^−^ (PANINFs-NO_3_^−^) ions (the chemical formulas are shown in Figure 4). Several variations of electrodes were prepared using the polymer conductor—it was used as a suspension dropped directly onto the surface of the inner electrode, which was GCE (glass carbon electrode), and as a component of a membrane cocktail, where it occurred in different amounts (0.5%, 1%, 2%). All polyaniline nanofiber-modified electrodes had better parameters than the unmodified electrode. Electrodes with PANI nanofibers doped with Cl^−^ ions had a wider range of linearity and a lower detection limit than those where PANINFs were doped with NO_3_^−^. On the other hand, the second ones exhibited a slope closer to the Nernst slope of −57.8 mV/decade. ISEs with both types of nanofibers did not show potential sensitivity to changes in pH in a very wide range. Furthermore, these types of electrodes showed excellent stability and potential reversibility and also had a lower membrane resistance value and a higher value of double-layer electrical capacitance relative to the unmodified electrode. These electrodes were successfully used to determine nitrate ions in real environmental samples, i.e., drinking, river and groundwater.

The same type of ionophore was also presented in the article [27], where a miniaturized version of the ion-selective electrode was presented in which the solid contact was mesoporous black carbon (MCB) and the conductive electrode was a silver wire placed in a glass capillary. MCB provides good ion-electron conductivity and, in addition, has a large specific surface. An additional advantage is its high availability and low price. The addition of MCB significantly stabilized the electrode’s response, ensuring adequate double-layer capacitance and decreasing the resistance more than a hundred times over coated wire electrodes. In comparison to electrodes described in the work of [26], these ISEs had a lower slope of −54.8 mV/decade, an order of magnitude more narrow range of linearity and a similar detection limit. Undoubtedly, the disadvantage of this electrode is its relatively short lifetime oscillating around 20 days. By comparing the electrodes proposed in the publications [26,27], it is clearly preferable to incorporate modifications such as PANINFs rather than MCBs.

Also, in the paper [28], for the determination of nitrate ions, an electrode in which TDMAN performed the function of the ionophore was proposed. The inner electrode was a screen-printed carbon electrode (SPCE), on whose surface cobalt (II, III) oxide nanoparticles (Co_3_O_4_NPs) were deposited, and acted as an ion-to-electron transducer. The solid contact was examined in terms of morphology and physical properties using XRD, EDS (energy-dispersive X-ray spectroscopy), SEM (scanning electron microscope) and TEM (transmission electron microscopy). Novelization of the electrode structure with hydrophobic Co_3_O_4_ nanoparticles prevented the formation of an aqueous layer and improved its response in relation to ISE without a Co_3_O_4_NP layer. This electrode also exhibited a good slope of −56.8 mV/decade, as well as a low limit of detection of 1.04 × 10^−8^ M. According to the EIS spectra, a reduction of membrane resistance was observed for the electrode modified with oxide nanoparticles. As was confirmed by comparing the results of SC-ISEs and spectrophotometer determinations, this electrode is certainly suitable for the analysis of various types of water samples.

The subsequent two electrodes were based on a glassy carbon electrode using polypyrrole (PPy) doped with NO_3_^−^ ions as the ion carrier. The electrodes differed in their solid contact layer, which, in the case of the electrode presented in the publication [29], was a pericarpium granati-derived biochar (PGCP) activated with phosphoric acid. PGCP was applied together with PPy, forming a bilayer membrane. Based on morphological studies PGCP was found to have a porous structure and, on the basis of the EIS measurements, also a low resistance, which translates into the possibility of using it as a transducer material for the charge transfer. In the second article, a double-layer structure consisting of gold nanoparticles (AuNPs) (solid contact) and a conductive polymer PPy-NO_3_^−^ (ion-selective membrane) was successfully used [30]. AuNPs solid contact layer and nitrate-doped polypyrrole molecularly imprinted polymer membranes were prepared by electrodeposition. Both of these electrodes had a slope that deviated from the ideal Nernstian slope and respectively amounted to −50.86 mV/decade for PGCP-ISE and −50.4 mV/decade for AuNPs-ISE. For the PGCP-modified electrode, the linearity range was in the range of 1.0 × 10^−5^–5.0 × 10^−1^ M and the LOD was 4.64 × 10^−6^ M, which is very similar to the results obtained for the gold nanoparticle-modified electrode, where the LOD was an order of magnitude higher. A positive aspect of the electrode, with solid contact in the form of AuNPs, is the favorable result of the aqueous layer test because no potential drift towards higher values of potentials was observed for this electrode compared to the unmodified AuNPs electrode. Finally, both the PGCP- and AuNPs-modified electrodes exhibited excellent long- and short-term stability and significantly better electrical parameters than the unmodified electrode. The PGCP-ISE was also successfully used for the determination of nitrates in samples from Shenzhen OCT wetland and laboratory wastewater with 4% RSD.

In publications [31,32,33] are presented electrodes that were connected by the presence of the same ionophore, which was nitrate ionophore VI. The ISE described by [31] was developed using screen-printing technology by applying inkjet printing gold onto the substrate, thereby forming a screen-printed gold electrode. Afterwards, the entire sensor was secured with Teflon tape and the membrane containing the ionophore was applied. The parameters of this potentiometric sensor were determined, i.e., the slope of the characteristic −54.1 ± 2.1 mV/decade and the linearity range between 1.0 × 10^−5^ M and 1.0 × 10^−1^ M. The water layer test was carried out with positive results. Unfortunately, the LOD for the obtained electrode was not included in the publication, and the electrical parameters were not determined.

Another article [33] described an SC-ISE that was produced by applying a membrane mixture containing varying amounts of PTFE (poly(tethrafluoroethylene)) (0%, 2.5%, 5%. 7.5%, 10%) onto a screen-printed electrode. The best results were obtained for an ISE in which PTFE constituted 5% of the membrane cocktail. This electrode was characterized by a slope of −58.0 mV/decade, which unfortunately reduced to −35.0 mV/decade after 20 days of measurements, which means that the lifetime of this electrode is not long, and further studies are required to eliminate this drawback. 

The last discussed electrode using nitrate ionophore VI is an SC-ISE based on the gold electrode (AuE) and thiol-functionalized reduced graphene oxide (TRGO) as solid contact material, which was used for the first time in this role [32]. This electrode was characterized by an almost Nernst response as the slope equaled −60.0 ± 0.5 mV/decade. The range of linearity of the described ISE was similar to electrodes possessing the same type of ionophore. An additional advantage of this SC-ISE is the low detection limit reaching micromole values and the wide working range of pH 2.0–10.0. Moreover, the use of TRGO improved the potential reversibility, which indicates that reduced graphene oxide performs well in the solid contact function due to its good conductivity. In addition, according to the EIS tests, the membrane resistance values of the electrode with TRGO are lower than those for the unmodified electrode. The aqueous layer test came off satisfactorily as no potential drift was observed. After improving the stability of the electrode in real samples, it will be possible to use it for the determination of ions in blood samples.

A new ionophore as the active ingredient in an ion-selective membrane used in the construction of nitrate(V) ion-selective electrodes was proposed by our group in the publication [34]. In the construction of SC-ISEs, a cobalt(II) complex with 4,7-diphenyl-1,10-phenanthroline (Co(Bphen)_2_(NO_3_)_2_(H_2_O)_2_) was used, which performed very well in this role and cooperated with the Ag|AgCl inner electrode. Trihexyltetradecylphosphonium chloride was selected as the ionic component of the membrane, which provided a constant concentration of chloride ions and represented a reversible redox system with the internal electrode Ag/AgCl/Cl-(silver/silver chloride electrode/chloride). A satisfying electrode response was obtained for this ISE, which was characterized by a slope of −56.34 mV/decade, a linearity range of 1.0 × 10^−5^–1.0 × 10^−1^ M and an LOD lower than the micromole value. The electrode exhibited a constant potential over a wide pH range of 5.4–10.6. Furthermore, the reversibility of the potential and its drift were at a very satisfactory level. This electrode was successfully applied to the determination of the concentration of NO_3_^−^ ions in tap, mineral and river water samples with reproducibility close to 100%. The same ionophore was used in an ion-selective electrode based on a glassy carbon electrode [35]. A nanocomposite consisting of multi-walled carbon nanotubes and the ionic liquid trihexyltetradecylphosphonium chloride (THTDPCl) was proposed as a solid contact and used as a membrane component. Various composites obtained from MWCNTs differing in structure were tested. The electrode with a nanocomposite containing nanotubes characterized by the highest porosity and homogeneity of the structure showed the best performance. In comparison to the previous electrodes with the same ionophore [34] based on Ag/AgCl/Cl, the electrode with a nanocomposite achieved a better slope closer to Nernst’s value, which was −57.1 mV/decade, a lower detection limit of 5.0 × 10^−7^ M and a linearity range that was an order of magnitude larger, as well as a wider working pH range. Interestingly, the addition of the nanocomposite directly to the membrane did not cause the redox sensitivity of the electrodes. Moreover, this simultaneously increased their hydrophobicity, so that no water film formation was observed between the membrane and the inner electrode in these SC-ISEs.

The ionophore in the form of TDANO_3_ (tetradecylammonium nitrate) was used both in a nitrate ion-selective electrode with a solid contact [36] and a liquid contact [37]. In the solid-contact electrode, the leading electrode was a screen-printed carbon electrode, and the solid-contact layer was a reduced graphite oxide aerogel (rGOA) [36]. A Nernst response of −59.1 mV/decade, a linearity range of 0.1 M to micromoles and a detection limit of 7.59 × 10^−7^ M were obtained. With success, this electrode was used for the determination of NO_3_^−^ ions in perilla leaves, where the results achieved by ISE were compared with chromatographic results. In the case of liquid contact, the inner electrode was one of the more commonly selected Ag|AgCl electrodes and the inner electrolyte—solution of 0.01 M KNO_3_ or 0.001 M KCl [37]. There is no doubt that the team carrying out the study on SC-ISEs succeeded in improving the parameters obtained for the competing electrode with LC. The slope of the characteristic differed from the theoretical value and was −53.7 ± 0.4 mV/decade, and, additionally, the values for linearity range and LOD were less satisfactory, so in this case the solid-contact ISE worked much more effectively.

In order to improve the performance of nitrate ion-selective electrodes with solid contact, the application of a nitron–nitrate complex (Nit^+^/NO_3_^−^) that acted as an ionophore was proposed [38]. A novel form of an ion-selective membrane that had the shape of a ‘sandwich membrane’ (bilayer membrane) was presented. This type of membrane was formed by pressing together two previously dried membranes, where one of them contained an ionophore and the other did not. The membrane prepared in this way was then applied onto the surface of the glassy carbon electrode with the non-ionophore side, where the part with the ionophore had direct contact with the testing sample. Multi-walled carbon nanotubes were used as the solid contact. The presented electrode showed a very wide linearity range of 8.0 × 10^−8^–1.0 × 10^−1^ M, and a low LOD of 2.8 × 10^−8^ M. The response of the GCE/MWCNTs/NO_3_^−^-ISM electrode was equal to −55.1 ± 2.1 mV/dec. The working pH range of such an electrode was between 3.5 and 10.0 pH and the lifetime was 8 weeks. The proposed ISE exhibited a very high tolerance to the presence of interfering ions, as we could observe from the rather low values of the selectivity coefficients. However, the authors did not report a value of this parameter for highly lipophilic ions, i.e., ClO_4_^−^ or SCN^−^, which are well-known interferents for nitrate electrodes. The ISE was successfully used to measure the concentration of nitrate ions in wastewater samples.

The article [39] presents liquid-contact nitrate ion-selective electrodes used for the determination of nitrates in hydroponic solutions. A design with two internal solutions and two membranes was used. The internal electrode was a silver chloride electrode placed in a PP tube filled with 0.1 M LiCl, which ended with a Nafion membrane—this constituted part 1. Part 1 was immersed in part 2, which was a second PP tube that ended with a PVC membrane, which, in turn, contained a 0.1 M LiNO_3_ solution (Figure 5). The ionophore included in the PVC membrane was THANO_3_ (tetraheptylammonium nitrate), which had already been used in nitrate ion-selective electrodes. On the basis of the calibration curve, a slope of −53.3 ± 0.1 mV/decade was determined, which deviates slightly from the ideal value for monovalent ions. The range of linearity and the detection limit took on average values and did not stand out from other nitrate electrodes. It was concluded that this type of electrode could be used in studies of NO_3_^−^ ions for more than four weeks in order to determine their concentration in hydroponic solutions.

The paper [40] presents a new form of solid contact in the form of TTF-TCNQ (tetrahiafulvalene-tetracyanoquinodimethane) illustrated in Figure 6. Here, the inner electrode was a glassy carbon disc onto which an intermediate layer was applied, which was followed by an ion-selective membrane containing an active ingredient, which was a nitrate ionophore V. The electrode showed great potential stability and was not sensitive to changes in the redox potential. The response of the electrode was almost Nernstian and amounted to −58.47 mV/decade, while the LOD was equal to 1.6 × 10^−6^ M. The presence of the proposed solid contact improved the selectivity of the nitrate electrodes and their electrical parameters, i.e., decreased the membrane resistance and increased the electrical capacitance. 

A comparison of the analytical parameters of various nitrate electrodes is presented in Table 1.

## 5. Ion-Selective Electrodes Sensitive to Fluoride Ions

Excessive amounts of fluoride ions become a threat not only to plant and animal organisms but also to humans. Increased content of this ion can cause disturbances in the normal sequence of the biological chain, and as a result, the ecological balance will be upset. In order to check how an excess of fluoride ions affects organisms, a number of studies were carried out over several years, which confirmed its neurotoxicity. High quantities of F^−^ ions damage cell organelles, i.e., mitochondria, and possess a mutagenic function, which in turn can lead to changes in gene expression. Furthermore, due to the ubiquitous presence of fluoride in drinking water and the large-scale use of fluoride, e.g., in toothpaste, it is necessary to monitor its concentration in the surrounding environment [41,42]. Consequently, as potentiometry is a rather cheap, fast and accurate method, it can be used to monitor fluoride ion concentrations in different types of samples. Over the past five years, 13 fluoride ion-selective electrodes have been proposed, and their design has been continuously improved to obtain better results.

The article [43] describes a new ion-selective electrode based on an electrode constituted by a titanium film, where lanthanum fluoride (LaF_3_) nanocrystals doped with europium (Eu) were proposed as the ion-conducting layer to improve the conductivity of the membrane. No solid contact intermediate layer was used here. The results obtained for this electrode were average, which was reflected in a typical linearity range of 1.0 × 10^−5^–1.0 × 10^−1^ M and a slope of −56 mV/decade. The deviation from the slope measurements was quite large and as high as −13 mV/decade. It is worth mentioning that the addition of Eu increased the slope of the characteristic by −10 mV/decade. In addition, it was possible to obtain LOD results at the micromole level. The disadvantage of these electrodes is the loss during the measurement of some Eu-doped LaF_3_ nanocrystals, which is the reason for the instability and non-repeatability of the potential.

The paper [44] presents a comparison of the performance of fluoride electrodes, each of which had a LaF_3_ single crystal membrane as an ion carrier. Two of the tested electrodes had the same silver inner electrode (AgE) to which different solid contact layers were applied. In the first case, PEDOT was used as the SC, which performed well in this role and undoubtedly contributed to stabilizing the electrode potential. It showed a slope of −56.0 ± 0.9 mV/decade. The second SC-ISE had an intermediate layer in the form of Ag paste. A super-normal slope was obtained, but the detection limit was the worst of all the fluoride electrodes described, with a slope of 2.0 × 10^−2^ M. Both SC-ISEs had the same linearity range. The third fluoride electrode presented in this article was a liquid-contact electrode, where the IE was Ag|AgCl and the internal solution was a mixture of phosphate buffer solution (PBS)—0.01 M Na_2_HPO_3_ and 0.02 M KH_2_PO_3_. For this electrode, the response slope of −38.6 ± 9.1 mV/decade was unsatisfactory and far from the book value. The best performing of the three presented electrodes was the PEDOT-modified SC-ISE, which had not only the lowest potential drift of 0.06 mV/min but also the highest electrical capacitance of 937 μF, which is 70 times higher than the Ag paste-modified electrode.

Novel fluoride electrodes with a LaF_3_ single crystalline membrane and different types of intermediate layers (solid contact or electrolyte solution) with the addition of Fe_x_O_y_ nanoparticles were presented in [45]. The intermediate layer consisted of Fe_x_O_y_NPs, which were incorporated into a membrane. For the SC-ISE, the internal electrode was a stainless-steel disc (SSDE), and for the liquid contact electrode, it was a silver chloride electrode, which was immersed in an electrolyte that was a mixture of KCl, HCl, 0.1 M KNO_3_. The working pH range was the same for both types of electrodes. The LC-ISE had a lower detection limit of 7.4 × 10^−8^ M and an order of magnitude wider range of linearity compared to the SC-ISEs, which in turn had a higher slope response. An additional advantage of the LC electrode was the high stability of the sensitivity over a period of two years.

A novel construction of a wearable spandex textile-based solid-contact fluoride sensor is presented in the article [46]. The scientific team proposed an SC-ISE based on a screen-printed carbon electrode (screen-printing spandex electrode) to which an intermediate layer of MWCNTs-COOH (carboxyl-functionalized multi-walled carbon nanotubes) was deposited, followed by a membrane containing the fluoride ionophore bis(fluorodioctylstannyl)methane shown in Figure 7. This electrode exhibited good reproducibility, stability and potential reversibility. The achieved Nernst characteristic response of −59.2 mV/decade and the nanomole value of the detection limit only confirmed that this sensor is suitable for the determination of even trace concentrations of fluoride ions. An additional advantage of this electrode is its high selectivity. An SC-ISE of this type can be used for the determination of either chemicals, biologicals or DFP (diisopropyl fluorophosphate).

Another electrode proposed for the determination of fluoride ions is SC-ISE, which in its design had a crystalline membrane consisting of a single crystal cadmium(II) Schiff base complex (CdLI_2_—cadmium iodide complex) (Figure 8) [47]. The electrode was obtained by impregnating the carbon paste electrode (CPE) with the Schiff ligand (E)-N_1_-(2-nitrobenzylidene)-N_2_-(2-((E)-(2-nitrobenzylidene)amino)ethyl)ethane-1,2-diamine(L) and its complex CdI_2_ (CdLI_2_). CdLI_2_ performed of the function of fixed-charge carriers. The presented electrode exhibited a good membrane resistance determined by the EIS method, a satisfactory slope of −58.9 mV/decade and a rather low detection limit of 1.2 × 10^−7^ M. In addition, very good selectivity against the ions such as IO_4_^−^, SCN^−^, SO_4_^2−^, CN^−^, ClO_3_^−^, Br^−^, Cl^−^ and I^−^ was recorded.

Fluoride solid contact electrodes with a new type of ionophore belonging to Lewis acidic organo-antimony(V) compounds were presented in [48]. A silver chloride electrode was used as the internal electrode, and a solution containing 0.2 M Gly/H_2_PO_4_ buffer and 0.001 M NaF was used as the internal electrolyte. All four ionophores are shown in Figure 9. The electrode in which tetrakis-(pentafluorophenyl)stibonium (Figure 9b) was used had the most Nernstian electrode response of −59.2 mV/decade and the lowest detection limit of 5.0 × 10^−6^ M. The electrode in which tetraphenylstibonium fluoride (Ph_4_SbF) (Figure 9a) acted as the ionophore had a slightly lower slope and higher LOD. The ionophore shown in Figure 9d was a component of the LC-ISE membrane cocktail with a slope of −57.8 mV/decade. All three electrodes had the same linearity range of 1.0 × 10^−5^–1 × 10^−1^ M. The electrode using tetrachloro-substituted organoantimony(V) (Figure 9c) had the weakest performance of all the LC-ISEs described in this publication. This ISE showed the lowest slope, the narrowest linearity range and the highest detection limit. The difficulty in carrying out measurements with these electrodes is that the pH of the samples must be within the range of three, as this is the most optimal value, so the acidity of the solution is recommended. These electrodes can be used to measure F^−^ ions, for example, in tap water.

A comparison of the analytical parameters of various fluoride electrodes is presented in Table 2. 

## 6. Ion-Selective Electrodes Sensitive to Chloride and Perchloride Ions

Chloride anions are quite common ions in drinking, ground and surface waters. Not only they are responsible for their saltiness, but they also affect the physiology of plants. Chlorine-containing compounds are used in the production of food or fertilizers. Chlorine is an important component for the human organism, and it is not toxic in small quantities, but the excess can cause, e.g., hyper-chloremia, which results in dehydration, diarrhea and metabolic problems, which, in turn, increases blood pressure and can lead to the damage of certain organs, e.g., the kidneys [49,50]. Perchlorate anions are very widespread in the world around us. They are present not only in various types of water, including ground and surface water, but also in plants and soil. In addition, they are used on a large scale in the industry for the production of explosives and pyrotechnics, and their increasing growth in the environment is linked to intensified rocket testing as perchlorates are used in rocket engine fuels. One of the main effects of overexposure to perchlorate ions is reported to be the disruption of the thyroid gland, due to the replacement of iodine by perchlorate, caused by a very similar ionic radius [51,52]. This makes it very necessary to monitor these ions. Other techniques are already used for this purpose, i.e., chromatography and spectroscopy, but potentiometry is an equally good and inexpensive way to determine the concentration of chloride and perchlorate anions. Since 2018, 16 Cl^−^-ISEs have been proposed with the use of novel charge transfer mediating layers and three leading ionophores (TDMACl, chloride ionophore(III), chloride ionophore(I)) and various types of nanocomposites (Figure 10). Whereas for the ClO_4_^−^ anions, three new SC-ISEs with different types of compounds used as an ionophore for the first time are presented.

In 2022, a paper that introduced novel Cl^−^-SC-ISEs was published. In the role of solid contact, the polyaniline nanofibers doped with Cl^−^ ions, multi-walled carbon nanotubes and three types of nanocomposites with different ratios of constituents that were PANINFs-Cl- and MWCNTs (2:1, 1:1, 1:2) were used [53]. The ionophore that was used was chloride ionophore(III) (Figure 10c), and the solid contact was applied by dropping an appropriate volume of the components directly onto the glassy carbon electrode. The proposed intermediate layers were investigated both from a morphological point of view, where their structure was presented on SEM images, and in terms of electrical properties that were determined by chronopotentiometry and EIS measurements. On the basis of these studies, the intermediate layer capacitance and resistance were determined, and the best results were obtained for a nanocomposite with a component ratio of 2:1, where C (electric capacitance) = 7.16 mF and R (resistance) = 0.21 kΩ. The same measurements were also carried out with the SC-ISEs after application of the membrane and again the best electrical performance was represented by the electrode containing this composite as an intermediate layer. All of the modified electrodes showed the same range of linearity and very similar LOD values of the order of 10^−6^ M and slopes close to the Nernstian value. Another advantage of these ISEs is their insensitivity to environmental changes, i.e., the influence of light or the presence of gases (O_2_ and CO_2_). The electrodes also exhibited very good selectivity, which makes them good devices for determining the concentration of Cl^−^ ions. This is confirmed by the determination of chloride anions using the proposed ISE, whose results were compared with those obtained using the classic Mohr method.

The next solution implemented in chloride potentiometric sensors is the use of an ion exchanger in the form of an ionophore, which is TDMACl (Figure 10a). This method was used, among others, by Kalayci [54] in a liquid contact electrode and by Pięk et al. [55] in an ISE with solid contact. In the Cl^−^-LC-ISE, a classical design of this type of electrode was used, where the IE was Ag|AgCl. The performance of this electrode was not outstanding and was average compared to the other electrodes listed in Table 3. On the other hand, in chloride SC-ISEs, novel conductive materials (redox mediators) belonging to the group of molecular organic materials (MOMs), i.e., TTF (tetrathiafulvalene), the chloride salt TTFCl and TTF-TCNQ (tetrathia-fulvalene-tetracyanoquinodimethane), as well as a combination of these materials with carbon black (CB) showing good hydrophobicity, conductivity and high specific surface area, were used as solid contacts. The modification of the electrodes improved their linearity range and slightly increased their slope but had little effect on LOD. Nanocomposites of CBs and MOMs turned out to be the best solid contact material as they exhibited better electrical performance, i.e., lower resistance and higher double-layer capacitance, and had a slightly better electrode response compared to electrodes based on molecular organic materials’ SCs. An additional advantage of the presented electrodes is their good selectivity and potential reversibility.

The paper [56] describes the novelization of SC-ISE with a new layer providing sensitivity to chloride ions. For this purpose, the carbon paste electrode (CPE) was modified with an ion-sensitive layer, which was a composite of graphitic carbon nitride (g-C_3_N_4_) that was anchored to a crystalline AgCl structure. The obtained structure was examined in terms of physical and morphological properties using techniques such as XRD, SEM and FTIR. After measurements, the best option was found to be the modification of the CPE with the addition of a 5% g-C_3_N_4_/AgCl composite. For this electrode, the parameters that were determined showed very good performance compared to the other Cl^−^-ISEs. The detection limit was at the lowest level compared to all the chloride electrodes presented in Table 3. A further advantage of this electrode is its very fast time of response and long-term potential stability (more than two months), as well as its good selectivity in the presence of interfering ions, i.e., I^−^, Br^−^ and CN^−^.

A comparison of the analytical parameters of various perchlorate electrodes is presented in Table 4.

Two kinds of electrodes differing in the type of layer providing sensitivity to chloride ions were reported in [57]. One of the electrodes had an ion-selective membrane with chloride ionophore(I) (Figure 10b), which was applied to an Ag|AgCl electrode, while the other ISE was a classical silver chloride electrode without an ionophore and ISM. It turned out that the electrode without the ion-selective membrane showed better stability and reproducibility of potential and kept a constant potential in solutions containing interfering ions than the proposed SC-ISE. Such an electrode can be used for the determination of chloride in human sweat.

Another example of chloride electrodes is the new SC-ISEs that have a membrane enriched with an AgCl:Ag_2_S:PTFE nanocomposite (two copies with different component ratios (1:1:2 and 2:1:2)), which provided selectivity of the electrode [58]. Additionally, nanoparticles of metal oxides, i.e., zinc oxide II (ZnONPs) or iron oxide (Fe_x_O_y_NPs), were used in the electrode construction as layers supporting charge transfer. These oxides were investigated using FTIR. In Table 3, the ISEs with the best analytical performance of the electrodes depending on the type of used oxide are compared. Iron oxides proved to be the best option for the studied electrodes. Despite the modifications, the slope of the produced electrodes was quite low and deviated from the theoretical value of −59.16 mV/decade, which undoubtedly indicates that the sensitivity of the proposed electrodes was quite weak.

One of three electrodes used for the determination of perchlorate ions was an all-solid-state coated wire electrode (CWE), where a platinum wire was immersed in a membrane cocktail to apply a membrane (opaque membrane—dixanthylium dye) [59]. The structure of the new ionophore is shown in Figure 11. The proposed electrode had a wide range of linearity, a fairly low detection limit of 5.0 × 10^−7^ M and an excellent working pH range of 1.5–11.0. The advantage of this ISE is a fast response time oscillating around 4 s and good selectivity towards many interfering ions. This electrode has been successfully used in the determination of perchlorate ions in the samples of mineral and tap water.

The next two SC-ISEs had a GCE as the basic electrode. In the first one, PEDOT was used as an intermediate layer and a dodecabenzylbambus[6]uril (Bn_12_BU [6]) as an ionophore (Figure 12) [60]. The proposed selectophore binds perchlorate ions very well due to an almost perfect match between the ion size and the receptor hole. Potentiometric, cyclic voltammetry, chronopotentiometry and electrochemical impedance spectroscopy measurements were carried out to verify if the proposed membrane component succeeds in its role. On the basis of the ISE response, the slope was determined, which equaled −59.9 mV/decade for the best electrode, which shows that this electrode has excellent sensitivity, a six-order linearity range and an LOD of 1.0 × 10^−6^ M. SC-ISE showed good stability and selectivity towards inorganic ions such as Br^−^, Cl^−^ and NO_3_^−^.

The second electrode with a GCE was the ISE with a solid contact in the form of single-walled carbon nanotubes (SWCNTs), where indium(III) 5,10,15,20-(tetraphenyl) porphyrin chloride (In^III^-porph) (Figure 13) was responsible for the selectivity towards the main ion [61]. The selectivity of the described electrode was at a high level and the analytical parameters of the ISE were very close to the SC-ISE proposed by Babaei et al. The electrode showed very good short-term stability and satisfying electrical parameters, e.g., the electrical capacitance equal to 27.6 ± 0.7 µF. The ClO_4_^−^-ISE response time is less than 10 s, and the lifetime oscillates around 8 weeks, which is not a very long period of time. The electrode was used for the determination of ClO_4_^−^ ions in firework samples, and the results were compared with those obtained by ion chromatography with satisfactory accuracy. In addition, the measurement of perchlorate anions in urea perchlorate, hydrazine, ethylenediamine and ammonium was also successful.

## 7. Bromide Ion-Selective Electrodes

Bromine is a fairly common element that occurs in large quantities in the biosphere. More often than in its free state, it is found in a bound form, e.g., in inorganic salts, as a decomposition product of hydrocarbons saturated with bromine, e.g., methyl bromide, which is a pesticide with toxic effects on humans and the rest of the ecosystem [62]. As bromine belongs to the group of halides, it has similar properties to chlorine or iodine. This is quite dangerous because in human organisms the substitution of iodine by bromide ions can occur, for example, in thyroid cells, which can cause dysfunction of this organ and of other functioning on the basis of thyroid hormone homeostasis. In addition, bromine as a toxin causes nervous system and neuropsychiatric disorders, problems with excessive muscle tremors and induces dermatological diseases, i.e., dermatitis and rashes [63,64]. Due to its toxicity, it is necessary to determine this element, e.g., in water and food, in order to verify whether the potential product can be safely consumed by humans. One of the best solutions involves the use of cheap and easy-to-use potentiometric sensors that selectively and accurately determine the concentration of this ion in samples. Several Br-ISEs have been proposed in the last five years, two of those described below have a SC and three have a LC.

One of the discussed liquid contact electrodes is a classical ISE with a silver chloride electrode, which was filled with 0.5 M NaBr solution [54]. The role of the bromide ion carrier was played by TDMABr (tridodecylmethylammonium bromide). This electrode was different from other Br^−^-ISEs because of its very wide working pH range of 1.0–11.0. The sensitivity of this electrode was remarkably high and the measurement results in the real sample were almost identical to those performed by ion chromatography. Another ISE with the same Ag|AgCl internal electrode is the electrode where Pt(II) 5,10,15,20-tetra(4-methoxyphenyl)-porphyrin (PtTMeOPP) was the selective ion carrier (Figure 14) [65]. In order to characterize the newly developed ionophore, both ^1^H-NMR and UV-Vis analyses were performed. A significant advantage of this electrode is its good selectivity and high sensitivity, as the slope of the response curve reached −64.4 mV/decade. It was successfully applied to the determination of Br^−^ ions in drug samples.

A two orders of magnitude wider range of linearity and a lower detection limit of 7.1 × 10^−8^ M was observed for LC-ISE in which the membrane was a core-shell nanocomposite based on boron-doped graphene oxide-aluminium fumarate metal organic framework (BGO/AlFu MOF) (Figure 15) [66]. The nanocomposite was characterized by XRD, FTIR and SEM techniques, and its absorption properties were investigated by UV-Vis spectroscopy technique. The response time of the proposed electrode was about 13 s and the obtained slope equaled −54.5 mV/decade, which is the worst value among all described Br^−^-LC-ISEs.

In the subsequent article [67] three variants of bromide electrodes with a solid contact, represented by POT sprinkled directly onto the glassy carbon electrode, were presented. The electrodes differed in the composition of the ion-selective membrane. The mesotetraphenylporphyrin manganese(III)-chloride complex (ionophore 1) was used as the ionophore in the first option and 4,5-dimethyl-3,6-diacetyl-o-phenylene-bis(mercuritrifluoroacetate) (ionophore 2) in the second. Membrane III contained in its composition, alongside DOS, ionophore 2 and PVC, the ion exchanger TDMACl. ISE I showed sensitivity to chloride ions; ISE with membrane III was rejected because it had significantly worse selectivity than ISE with membrane II. Electrode II, which exhibited the best performance, had a rather low detection limit of 2.0 × 10^−9^ M but a narrow linearity range of 1.0 × 10^−8^–1.0 × 10^−6^ M.

A comparison of the analytical parameters of various bromide electrodes is presented in Table 5. 

## 8. Iodide Ion-Selective Electrodes

Iodine is one of the most crucial elements for the human body. It not only influences the proper functioning of the neurological system but also regulates metabolic processes. It ensures the proper functioning of the thyroid gland and the development of bones or muscles. It is very important to supply the body with adequate amounts of this micronutrient via food and liquids. A deficiency or excess of this element leads to hypothyroidism or hyperthyroidism, which, in turn, causes a number of other health problems, such as circulatory disorders, weakness and mental illnesses [68,69,70]. Therefore, it is essential to monitor the amount of iodine in the human diet, and, for this purpose, potentiometric sensors characterized by selectivity and good sensitivity to iodine can be used, as well as other techniques. Thirteen ISEs are described below, which differed in the type of ionophore, the intermediate layer, the type of contact and the lead electrode.

For the determination of I^−^ ions, a new iodide electrode was proposed with an AgI:Ag_2_S:PTFE ion-sensitive membrane in a ratio of 1:1:2, which was enriched with nanoparticles of zinc oxide (ZnO) [71]. Several electrodes differing in the amount of ZnO in the range of 1–5 wt.% were fabricated. The best of the proposed membranes turned out to be the one marked as M2, which contained 10 mg ZnO (it was examined by SEM and EDS). Of all the LC-ISEs, this I^−^-ISE had the slope that was the closest to Nernstian’s. The range of linearity for the presented potentiometric sensor in comparison to measurements using voltammetric methods was actually wider by as much as two orders of magnitude. This electrode was successfully used for the determination of penicillin in pharmacological samples.

Classic electrodes with liquid contact and silver chloride internal electrodes are described in the works of [54,65,72]. They differed in the type of ion-sensitive component that provided good sensitivity towards iodide ions. Kalayci used tridodecyl-methylammonium iodide (TDMAI) for I^−^ ion capture. The performance of this electrode was not significantly different from other I^−^-LC-ISEs but was better than that for the sensor proposed by [65], where PtTMeOPP was used as the ionophore. This electrode had a lower slope of −52.3 ± 0.4 mV/decade, while the other parameters, i.e., linearity range or LOD, were not significantly different. A final example of LC-ISEs with Ag|AgCl as the internal electrode are electrodes modified with newly proposed active ingredients to provide selectivity to iodide ions. The first such material is the metallic complex platinum(IV) tetra-tertbutylphthalocyanine dichloride (Pc^t^PtCl_2_) (Figure 16a) [72]. In addition, three other composites consisting of the above-mentioned complex and the ionic liquid cetylpyridinim chloride (CPCl) (Figure 16d) (CPCl + Pc^t^PtCl_2_) or cetylpyridinuim bromide (CPBr) (Figure 16c) (CPBr + Pc^t^PtCl_2_) or 1,3-dicetylimidazolium iodide (DCImI) (Figure 16b) (DCImI + Pc^t^PtCl_2_) were used. These modifications did not have the desirable effect as these electrodes had the lowest detection limit and narrowest linearity range compared to the other cited LC-ISEs, and the slope of the characteristics was less than −50 mV/decade for all electrodes except the one modified with the CPBr + Pc^t^PtCl_2_ composite. Moreover, in the publication, electrodes with a solid contact that had the same active components in the ion-selective membrane as those mentioned above, i.e., Pc^t^PtCl_2_, CPCl + Pc^t^PtCl_2_, CPBr + Pc^t^PtCl_2_, DCImI + Pc^t^PtCl_2_, were also described. The parameters of the proposed SC-ISEs were clearly more satisfactory than those for the classic ISEs. The best results were obtained for electrodes modified with a composite of the metal complex and ionic liquid DCImI + Pc^t^PtCl_2_ and CPBr + Pc^t^PtCl_2_. The second one was used for the determination of iodide ion content in pharmaceuticals, i.e., Iodomarine 100 and Iodobalanse 100.

Two novel ionophores providing sensitivity to I^−^ ions were characterized in a publication [73]. They were used as a component of an ion-selective membrane in an SC-ISE, where a combination of SPE with PANI was used as a solid contact. The first of the new active ingredients is XB_1_ (tripodal halogen bonding (XB) ionophore) (Figure 17) and the second is HB_1_ (H-triazal analogue of XB_1_) (Figure 18). Both ionophores were investigated using NMR techniques (the effect of XB interactions between I^−^ ions and the ionophore was determined). It is also worth mentioning that the determined selectivity was incompatible for the SCN^−^ ion with the Hofmeister series, which indicates that the ionophore binds with the I^−^ ions, as a result of the NMR measurements. Among the two proposed structures, the SC-ISE with HB_2_ as the ionophore proved to be the better solution with higher sensitivity, but it was not significantly different from the competitive ISE with XB_1_ in the ISM structure.

A comparison of the analytical parameters of various iodide electrodes is presented in Table 6.

## 9. Ion-Selective Electrodes Sensitive to S^2−^, SO_3_^2−^ and SO_4_^2−^ Ions

Sulfur is one of the basic micronutrients that are essential for the human organism. It is not only a component of amino acids and proteins, but it is also an important element crucial and necessary for metabolic processes that take place in the body [74]. Sulfur is, apart from other things, a component of insulin, and a deficiency of this element can therefore lead to hyper- or hypoglycemia; it has an anti-inflammatory effect and is involved in the synthesis of connective tissue. Sulfur compounds are also important nutrients for the growth and development of plants [75]. Both a deficiency and an excess of this element, its compounds and ions are not beneficial to humans and animals, as well as to the flora and the rest of the environment. Therefore, a variety of techniques are used to determine the amount of these molecules. One of the leading methods is potentiometry, so newer and newer ISEs are being constructed to provide accurate and sensitive measurements. This review focuses on ten electrodes that are designed to monitor the concentration of sulfur-containing ions, where four are sensitive to S^2−^ ions, two to SO_3_^2−^ ions and four to SO_4_^2−^ ions.

For the detection of S^2−^ ions, four different ISEs were recently constructed, including three with a liquid contact and one with a layer of solid contact. One of the LC-ISEs described in the publication [76] had an ion-selective membrane made of Ag_2_S, which was obtained by sedimentation. The membrane in the form of a disk was placed at one end of a PVC case containing a 1 × 10^−6^ M Na_2_S (electrolyte solution), which provided a conductivity between the membrane and the Ag|AgCl inner electrode. The parameters that were obtained for the proposed electrode were quite satisfying, i.e., a low limit of detection of 2.3 × 10^−7^ M, a six-order linearity range and an almost Nernstian slope of −28.2 mV/decade. A fairly fast electrode response of 5–17 s was also obtained. This electrode can be used for the determination of sulfide ions in various types of solutions and industrial water used in oil refineries.

Another two LC-ISEs were described by Matveichuk et al. [77]. In this study, the electrodes differing in an ion carrier were compared. One of these was the previously used 4-(trifluoroacetyl)heptyl ester of benzoic acid (TFA-BAHE), and the other was the new PVC-modified 4-(trifluoroacetyl)benzoate (TFAB-PVC) shown in Figure 19. The use of TFAB-PVC resulted in an extension of the electrode lifetime in comparison to ISE, where TFABAHE was used. This was probably due to the fact that the covalent bonding between the TFAB and the PVC matrix prevented its elution from the membrane. Depending on the environment, this process has a different course. Both of the presented electrodes had almost identical parameters, i.e., slope, linearity range and LOD. The change of ion carrier had a positive effect on the electrode’s lifetime, and an additional advantage is the possibility to perform measurements in alkaline and acidic environments.

The last S^2−^-ISE is an electrode composed of a silver wire (inner electrode) to which a transducer layer in the form of reduced graphene sheets (RGSs) (solid contact) was electrodeposited by reducing graphene oxide [78]. The final step in the preparation of this electrode was the electrodeposition of Ag_2_S constituting the ISM. The proposed electrode showed good 7-day potential stability, as well as satisfactory selectivity, but had an order of magnitude more narrow linearity range than the other presented ISEs. The electrode exhibited a super-Nerstian slope of characteristics of about −200 mV/decade. The authors explain this by the phenomenon of ion bonding in the solid contact layer that results in increased transport of ions from the diffusion layer.

Another potentiometrically monitored anion is the SO_3_^2−^ ion, and the electrodes sensitive to it were described in the work [79]. Two screen-printed electrodes with a polymeric membrane containing cobalt(II) phthalocyanine (CoPC) as a material for selective detection of SO_3_^2−^ ions were proposed. These two electrodes differed in the solid contact layer, which was constituted by organic conductors, i.e., carboxyl functionalized multi-walled carbon nanotubes (MWCNTs-COOH) or polyaniline nanofibers (PANINFs). Both described electrodes had a rather narrow range of linearity. The MWCNTs-COOH-modified electrode had a better electrode response of −29.8 mV/decade and a lower LOD, as well as better electrical parameters of C = 26.1 μF and R = 5.3 kΩ compared to the one in which the intermediate layer was PANINFs. In addition, the ISEs were not sensitive to changes in environmental conditions, i.e., the presence of light and gases (O_2_ and CO_2_). They also positively passed the water layer test.

Among the four ISEs intended for the determination of SO_4_^2−^ ions, the CPE (carbon paste electrode) showed the best predisposition in this direction, where the function of the ionophore was performed by a Schiff base complex with nickel (Figure 20) [80]. To verify whether there are no sulfide ions in the complex and if it is suitable to be an ionophore, UV-Vis spectra were determined. Good parameter values were obtained: a Nernst response of −29.7 mV/decade, a nanomole LOD value of 5.0 × 10^−9^ M and a satisfactory linearity range of 7.5 × 10^−9^–1.5 × 10^−3^ M. The effect of temperature and the sensitivity of the potential to changes of pH were also tested. The electrode was successfully used for the determination of SO_4_^2−^ ions in real samples, i.e., mineral water or blood serum.

Three further electrodes that differed in the type of ion carrier were described in articles [77,81]. In all electrodes, a solution of 0.01 M Na_2_SO_4_ and 0.001 M KCl was used as the internal electrolyte. The ingredient providing selectivity in the electrode described in the paper [81] was the higher quaternary ammonium salt 3,4,5-tris(dodecyloxy)benzyl(oxyethyl)_3_trimethylammonium chloride ((oxyethyl)_3_TM) which was the ion exchanger (Figure 21). Meanwhile, in the work [77], the anionic ion carrier TFABAHE and the neutral ion carrier TFAB-PVC were used for this purpose. Among the above-mentioned electrodes, the highest slope of the characteristic of −27 mV/decade and the lowest limit of detection showed the ISE in which the ion exchange (oxyethyl)_3_TM was used as the ionophore. Furthermore, the lifetime of this electrode was approximately 1 month. The second in terms of quality was the TFABAHE-modified electrode, which slightly outperformed the competitive ISE with TFAB-PVC as the ion carrier.

The basic analytical parameters of ISEs sensitive to S^2−^, SO_3_^2−^ and SO_4_^2−^ ions are presented in Table 7.

## 10. Ion-Selective Electrodes Sensitive to Phosphates

Phosphorus ions are a fairly common constituent contained in food, drinking water and soil. They have a major impact not only on human health but also on the economy, flora, fauna and industry. As a micronutrient, phosphorus is involved in the synthesis of phospholipids and other nucleic proteins [82]. In addition, phosphorus compounds are used quite abundantly in the production of fertilizers. They permeate from the soil into the groundwater and then further, which results in surface eutrophication of water reservoirs [83,84]. This process reduces the oxygen saturation of water and causes the death of many aquatic organisms. Phosphorus ions are also present in processed foods, e.g., meats, beverages and vegetables, where they have a preservative function, but an excess of phosphorus can cause a negative response in the human body, i.e., circulatory problems or problems with the urological system, e.g., kidneys. The maximum number of phosphates in drinking water according to the. WHO is 1 mg/L [85]. For this reason, it is extremely important to improve the methods used to detect phosphate concentrations. One possibility is the use of ion-selective electrodes, which are not only a simple and inexpensive tool, but also give satisfactory, accurate results at a micromole level for phosphates. Over the past few years, four ISEs sensitive to H_2_PO_4_^−^ ions, seven selectively detecting HPO_4_^2−^ ions and three sensitive to PO_4_^3−^ ions have been developed.

One of the four electrodes used to determine the H_2_PO_4_^−^ ion was a type of ISE with an internal electrolyte. In the publication [86], a new ionophore 1,3-phenylenebis(methylene)[3-(N,N-diethyl)carbamoylpyridinium] hexafluorophosphate (bis-meta-NICO-PF_6_) (Figure 22) that provides selectivity for ISE was presented. The same type of ionophores was also presented in bis-, tris- and para-isomerism, but it was the meta-isomer that had the highest affinity for phosphates, which was confirmed by NMR studies. In this paper, two types of the applied membrane were presented: one of them was a classic polymeric membrane, while the other one was a double membrane consisting of a polymeric membrane coated with a silicon rubber (SR) solution layer that was in direct contact with the solution after evaporation of the solvent. The SR layer was introduced to prevent the washing-out of the ionophore from the membrane. Thanks to this, the life of the electrode was extended to approximately 40 days and the electrode response was improved by −15.4 mV/decade compared to an electrode without SR. An additional advantage of this ISE is its good selectivity.

The next three electrodes are electrodes without IE. They have, as leading electrodes, glassy carbon (GCE) [87], carbon paste electrode (CPE) [88] and Cu wire [89], respectively. The publication [87] presents a novel nanocomposite consisting of polypyrrole (PPy), cobalt and mesoporous ordered carbon (Co-PPY-OMC), which was deposited on GCE in a one-step electrodeposition at constant potential. In order to determine the appropriate component ratios of the selective layer, Response Surface Methodology (RSM) combined with Box Behnken Design (BBD) was used. EIS measurements were carried out and showed that the use of the nanocomposite resulted in a reduction of resistance. Moreover, a satisfactory LOD of 6.8 × 10^−6^ M and a fairly short response time of 9 s were observed. The ISE had a relatively low potential drift of 1.45 µV/s but unfortunately showed a variable slope depending on the pH value. On the other hand, in the paper [88], the preparation of the hydrogen phosphate ion-imprinted polymer nanoparticles (nano-IIP) in ACN/H2O (acetonitrile/water) with the help of a matrix in the form of H_2_PO_4_ was presented. Nano-IIP is a novel component providing the carbon paste ISEs selectivity to H_2_PO_4_ ions. The LOD of the presented electrode reached micromole values. In addition, the electrode is not resistant to pH changes and its potential varies with the pH value. It is associated with the deprotonation that occurred at the interface between the solution and the nano-IIP due to the presence of pyridinium groups. For both Co-PPY-OMC/ISEs and nano-IIP/ISEs the slope is quite weak, deviating fairly strongly from the Nernst value:−31.6 mV/decade and −30.6 mV/decade, respectively. This is quite a disadvantage of such ISEs and may be related, for instance, to the leaching of ionophores from the membranes. The electrode described in the publication [89] had a solid crystalline membrane obtained from a compressed mixture of Ba_3_PO_4_, Cu_2_S and Ag_2_S salts connected to a Cu wire. The electrode showed a much higher sensitivity because the slope was similar to Nernstian:−57 mV/decade. In addition, it had the widest linearity range of 1.0 × 10^−6^–1.0 × 10^−1^ M, as well as the lowest detection limit of 2.4 × 10^−7^ M. The slope value was determined for the electrode conditioned in 0.1 M NaH_2_PO_4_ buffer solution, while for the electrodes stored in air and water, the slope was −20.6 mV/decade and −27.5 mV/decade, respectively. Unfortunately, in the publication, there is no explicit statement as to which form of phosphate ion the electrode is sensitive to.

Among the HPO_4_^2−^ ion-selective electrodes, there are six electrodes with solid contact and two with an internal electrolyte. Two of the SC-type electrodes had a Cu wire as the leading electrode. In the carbon paste ISE construction, a new copolymer (whose structure is shown in Figure 23) and MWCNTs were used [90]. The newly proposed layer was characterized using FTIR, XPS, TG/DTG-DTA and SEM techniques. A satisfying response of the electrode for the divalent main ion of −30.7 ± 0.4 mV/decade was obtained. The ISE exhibited a potential drift of 0.48 mV. In addition, the lifetime of HPO_4_^2−^-ISE was 17 weeks, so it is possible to use it repeatably. The solid-state monohydrogen phosphate sensor was used to determine the content of HPO_4_^2−^ ions in many samples with very good reproducibility, i.e., tap water, dam water and river water. 

The article [91] presents a new SC-type electrode in which Bi particles were electrolytically deposited onto a platinum wire (Pt wire) from a solution of potassium citrate and bismuth using a Shoddy diode and a function generator. The ion-sensitive function was performed by a BiPO_4_ membrane, which was deposited by electroplating using chronoamperometry. The surface of the resulting structures was studied by the SEM technique, which confirmed that both materials formed a homogeneous layer. The described electrode was characterized by good analytical performance with a slope of −30.3 mV/decade, a six-order linearity range and an LOD equal to 7.7 × 10^−7^ M. Moreover, the advantage of the electrode is its response time of 1–2 s and also good reversibility and stability of the potential in different concentrations. The electrode showed a lifetime longer than 90 days. The most convenient working range of the pH value is between 5–9. The electrode was successfully applied to the determination of HPO_4_^2−^ ions in natural water. The disadvantage of this electrode is its high sensitivity to chloride ions, and, therefore, it cannot be used in chlorinated solutions, e.g., seawater samples, glacial water samples, etc.

A new electrode based on a molybdenum wire was also proposed for the determination of HPO_4_^2−^ ions [92]. The Mo wire was placed in a silicon tube in which one end was covered with a resin in order to prevent it from entering the sample solution. Subsequently, electrochemical deposition of MoO_2_ + PMo_12_O_40_^3−^ (molybdenum dioxide and molybdophosphate) was carried out until the electrode turned black and the solution turned green. This layer was the ion-sensitive layer towards the hydrogen phosphate ions. The measurements were performed in solutions of the main ion at different pH values, and pH = 9 was found to be the most optimal value. On the basis of the calibration curve, a slope of −27 mV/decade, an LOD equal to 1 μM as well as a linearity range of 1.0 × 10^−5^–1.0 × 10^−1^ M were determined. The disadvantage of this electrode is its rather long response time of approximately 5 min.

In the work [93], two kinds of hydrogen-phosphate electrodes in which two types of ion-selective membranes were used were proposed. Both electrodes contained a mixture of silver salts (Ag_3_PO_4_ + Ag_2_S) but differed in the dopants, the method of membrane preparation and the type of electrode material. In the electrode marked ‘type 1 membrane’, the inner electrode was a stainless steel disc (SSD) and the third membrane component was polytetrafluoroethylene (PTFE) constituting 50% of the membrane weight. The components of the mixture were pressed together then placed in a Teflon casing and connected to the SSD. In the electrode described as a ‘type 2 membrane’, the inner electrode was a copper wire while the salt mixture that was used was enriched with multi-walled carbon nanotubes (MWCNTs) (2%). The membrane was prepared by homogenizing the composite mixture with linseed oil, which was then placed in a plastic tube. Subsequently, the Cu wire was immersed in it and then allowed to dry for three days. Both electrodes had the same linearity range and a similar LOD but differed significantly in the slope, which was −21.0 mV/decade for the PTFE-modified electrode and −32.6 mV/decade for the ISE with MWCNTs, which was considerably better. For some parameters, the results were only presented for one electrode, i.e., the response time was only presented for the ISE ‘type 2 electrode’ and amounted to nearly 60 s. According to the authors’ description, the type 1 electrode showed a lifetime of two years, while the type 2 electrode showed a lifetime of only a few days. Definitely, a better comparison of the properties of the conductive materials would be if the same type of internal electrode and method of electrode formation was used because a comparison of the parameters of two completely different electrodes is not adequate.

Only two of the seven electrodes sensitive to HPO_4_^2−^ ions had a construction using an internal electrolyte; they are described in [77], where a new material performing the function of a neutral ion carrier was presented, which had already been used for the S^2−^ and SO_4_^2−^ ion-sensitive electrodes. In this case, ion-sensitive layers in the form of TFAB-PVC and TFABAHE worked similarly well, as can be seen by the good response and LOD of the electrodes. The TFAB-PVC-modified electrode showed a significantly longer lifetime, while the presence of TFABAHE reduced the lifetime of the electrodes by 60 days compared to the unmodified electrode with this carrier. The presence of chemically modified PVC improved the selectivity and limit of detection.

The last type of electrodes described was PO_4_^3−^-ISEs, which are presented in [94]. All three electrodes were made up of Cu wire immersed in an internal electrolyte of 0.001 M KCl + 0.001 M Na_3_PO_4_. To ensure selectivity, membranes were prepared using phosphate-imprinted polymers as ionophores: IIP-1 (chitosan-La(III)-PO_4_^3−^), IIP-2 (chitosan-La(III)-AAPTS-PO_4_^3−^) and IIP-3 (AAPTS-La(III)-PO_4_^3−^). The applied materials were investigated using FTIR. Calibration curves were obtained, based on which the slope, LOD and linearity range were determined. The slope of the characteristics was extremely different from Nernsian’s and equaled, respectively, −3.2 mV/decade, −1.9 mV/decade and −3.7 mV/decade for IIP-1/ISE, IIP-2/ISE and IIP-3/ISE, indicating their very weak sensitivity. Despite of the inclusion of information in the publication that each electrode had a linearity range of 1.0 × 10^−6^–1.0 × 10^−2^ M, this value is only true for IIP-1-ISE, while this value was incorrectly reported for the other ISEs. The electrodes showed high sensitivity to changes in the pH value and extremally long response times of 150, 130 and 30 min sequentially for the ISEs modified with IIP-1, IIP-2 and IIP-3. 

A comparison of the analytical parameters of various phosphate electrodes is presented in Table 8.

## 11. Ion-Selective Electrodes Sensitive to Tiocyanate Ions

The thiocyanate anions have toxic impacts on humans and other living organisms. One of the effects of the excessive amount of SCN^−^ ions, e.g., in milk, is the inhibition of iodine uptake by the thyroid gland, loss of consciousness and intense dizziness. Newborns and pregnant women are the most susceptible to the influence of SCN^−^, as well as organisms living in communities located far from saline water reservoirs, which are the source of iodine [95,96]. Increased thiocyanide concentrations in the human body are associated with excessive smoking (passive and direct smokers) or industrial pollution. The thiocyanates are the compounds that have the greatest environmental impact on iodine metabolism in the body and therefore on the occurrence of diseases associated with thyroid malfunction [97,98]. Consequently, it is necessary to monitor the concentration of these ions in order to protect ourselves from being excessively exposed to them. For this purpose, we can use low-cost and fast-measuring instruments such as ion-selective electrodes. Over the past few years, five new modifications of SCN^−^-ISEs have been proposed to improve their performance.

In the article [99], liquid contact electrodes in the form of a 0.01 M KSCN solution were used to determine SCN^−^ ions in the human saliva of smokers and non-smokers. Several ionophores for the detection of thiocyanides were described in the present study, but ultimately 3,4,5-tris(dodecyloxy)benzyltrilauryl ammonium (TL) bromide was used as the membrane component of the presented ISE (Figure 24). The electrode had a wide pH range of 0.5–12.5 and a good detection limit of 5.6 × 10^−6^ M. The ISE was also characterized by an average slope and linearity range.

Urbanowicz et al. presented three types of SCN^−^-ISEs, where one was a liquid and two were solid contact electrodes [100]. All three electrodes in the membrane had as an ionophore a tetrakis-(4-diphenylmethylphosphonium-butoxy)-tetrakis-p-tert-butylcalix [4]arene tetrathiocyanate (Figure 25). For the liquid-contact electrode, the ionophore membrane was placed in the Ag|AgCl electrode body and then immersed in 0.001 M KCl, which was the internal electrolyte. On the other hand, electrodes without an internal electrolyte were created by dropping an appropriate volume of membrane mixture on the surface contact of GC electrodes or Au rods. In this study, membranes with three plasticizers o-NPOE, BBPA and chloroparaffins were investigated. Among all three electrodes, the best performance was obtained for SCN^−^/GCE/ISE, where a Nernstian slope of −59.9 ± 0.3 mV/decade was achieved. Both SC-type electrodes showed an order of magnitude wider range of linearity with respect to LC-ISE. Analogous to the electrode described in the previous paper, the concentration of SCN^−^ ions in the saliva of non-smokers and smokers was investigated.

The last SCN^−^-ISE described in this review is the LC electrode (Ag|AgCl immersed in an electrolyte constituted by a mixture of 0.01 M NaSCN and 0.1 M NaCl) [101]. The ion-selective membrane was a sol-gel-based matrix in which the active ingredient was tricaprylylmethylammonium thiocyanate (Aliquat336-SCN) (ionophore). The results obtained for this electrode were compared with those for an ISE using a conventional polymeric PVC membrane. Better selectivity towards anions, i.e., ClO_4_^−^ and SiO_4_^−^, was obtained. Compared to the other liquid contact electrodes, it had the highest slope value (highest sensitivity), while the other parameters were very similar. This electrode was also used for the determination of SCN^−^ ions in human saliva.

A comparison of the analytical parameters of various thiocyanate electrodes is presented in Table 9.

## 12. Ion-Selective Electrodes Sensitive to Other Ions

The development of ion-selective electrodes is still in progress and newer and newer sensors that selectively detect a large range of ions were introduced by researchers. Over the past five years, single solutions have been presented for the determination of ions such as AsO_4_^3−^, BO_3_^3−^, CH_3_COO^−^, CO_3_^2−^ and SiO_3_^2−^.

An electrode sensitive to AsO_4_^3−^ ions was presented in 2018 by Khan et al. [102]. A new fibrous poly-methylmethacrylate-ZnO (PMMA-ZnO) ion exchanger (Figure 26) was proposed, which was prepared by a solid-gel method. The newly formed material was then characterized by techniques, i.e., FTIR, SEM, TEM, XRD, TGA and EDX. It showed very good properties and was successfully applied as an active component of an ion-selective membrane providing selectivity to arsenic ions. The presented LC-ISE exhibited a super-Nerstian response of the electrode (slope of −28.6 mV/decade for the trivalent ion), a nine-order linearity range and a nanomole detection limit of 1.0 × 10^−9^ M. The response time of such an electrode is about 35 s, whereas a great advantage is the possibility of using it for a period of 12 months.

In order to monitor glucose and glycate in blood and to determine the boron in real samples, an electrode sensitive to BO_3_^3−^ ions was invented [103]. This electrode is based on a composite of multi-walled carbon nanotubes and Ag_2_B_4_O_7_, which simultaneously acts as a solid contact layer and a BO_3_^3−^ ion trapping material. The MWCNTs provide good ionic conductivity, while the other component of the composite provides ISE selectivity. The composite membrane was placed in a tube and compressed before being placed on a Cu wire. The performance of the electrode was not remarkable because the LOD = 5.6 × 10^−5^ M and the linearity range was equal to 1.0 × 10^−4^–1.0 × 10^−1^ M. The advantage is that the electrode is not sensitive to the presence of interfering ions and its lifetime extends to 18 weeks. Moreover, good potential reversibility of such an electrode was reported. It was successfully used for the determination of boron in rocks, soil and water because comparable results to those achieved by the ICP-MS technique were obtained.

The determination of CH_3_COO^−^ ions in aqueous solutions was presented in [104], where a solid contact electrode with 1,3-bic(carbosyl)urea derivate acting as the ionophore was described (Figure 27). The function of the solid contact was performed by the conductive polymer PEDOT, which was deposited on the GCE surface by galvanostatic electropolymerization. A polymeric ion-selective membrane containing the proposed active ingredient was then dropped on the prepared surface. The impedance measurements using EIS as well as the parameters determined by potentiometric measurements were carried out for CH_3_COO^−^-ISE, all of which were not impressive. A narrow linearity range and a low LOD were obtained, as well as a slope = −51.3 mV/decade. Measurements cannot be performed in alkaline solutions due to the high probability of OH^−^ ions interference. In comparison to the electrode that contained the TDMACl ion exchanger, a weaker influence of interfering ions, i.e., SCN^−^, I^−^, Br^−^ and NO_3_^−^, was obtained.

The new carbonate electrode proposed by the team of Zhang et al. can be used for the exploration of deep-sea hydrothermal activity [105]. It has carbonate film as the solid contact and carbonate ionophore VII as the ionophore. In order to make such an electrode selective for CO_3_^2−^ ions, a carbon film was first applied electrochemically to the Ni wire and then that electrode was immersed in a solution containing the ion-sensitive component. To characterize the formed film on the electrode morphologically, the SEM technique was used. Subsequently, analytical parameters that were at a very good level were determined, including slope = −30.4 mV/decade and LOD = 2.8 × 10^−6^ M. The electrode also showed good reproducibility and potential stability, so the carbon film proved to work well as a solid contact.

To ensure the possibility of measuring silicates in aqueous solutions containing small amounts of chlorine, an ISE which was Ag wire coated by the Pb film was proposed [106]. Selectivity for silicates was provided by a PbSiO_3_ membrane, which was electrochemically applied to the surface of the sensor. The electrode was characterized by good sensitivity, as is evidenced by an over-Nernst slope for this type of ion, as well as a satisfying detection limit of 2.8 × 10^−6^ M. An advantage of this electrode is its fast response time of 5 s. It showed good selectivity towards SiO_3_^2−^ ions and was not affected by the interfering ions NO_3_^−^, CH_3_COO^−^ and CO_3_^2−^. The electrode needs to be improved in terms of its sensitivity to Cl^−^ ions so that it can be applied to different types of samples.

The basic analytical parameters of other ISEs described above are presented in Table 10.

## 13. Conclusions

This has been a review of the new materials used in the construction of ion-selective electrodes for anion determination. From this overview of more than 100 different anion-selective electrodes, two main research directions can be distinguished, one concerning the development of new ionophores and the other concerning the improvement of the construction, mainly related to the introduction of new materials of solid contact or components of paste electrodes. Depending on the type of ion, one or the other direction is dominant, e.g., in the case of nitrate electrodes, most of the papers concern solid contact materials, while in the co-publications concerning new phosphate electrodes, papers describing new ionophores predominate. Among the active substances of the membrane used to obtain the selectivity of a specific ion, ion exchange substances and sparingly soluble salts still dominate. Conductive materials used as intermediate layers or membrane modifiers in ASS-ISEs are mainly conductive polymers, carbon nanomaterials, metal nanoparticles and composite or hybrid materials.

The use of new materials in the construction of ISEs allowed to obtain electrodes with better performance and more convenient to use. This is a valuable achievement considering the numerous advantages of potentiometry, and it is all the more important that there are far fewer alternative determination methods for anions.

In the area of anion-selective electrodes, the development of effective ionophores for hydrophilic anions (carbonates, phosphates and sulfates) is still a current research topic. Another prospective direction of research includes the use of composite materials, especially nanocomposites, based on carbon nanomaterials and metal oxide nanoparticles. These materials combine the valuable properties of the constituent components, which opens up new fields for their effective applications.

## Figures and Tables

**Figure 1 materials-16-05779-f001:**
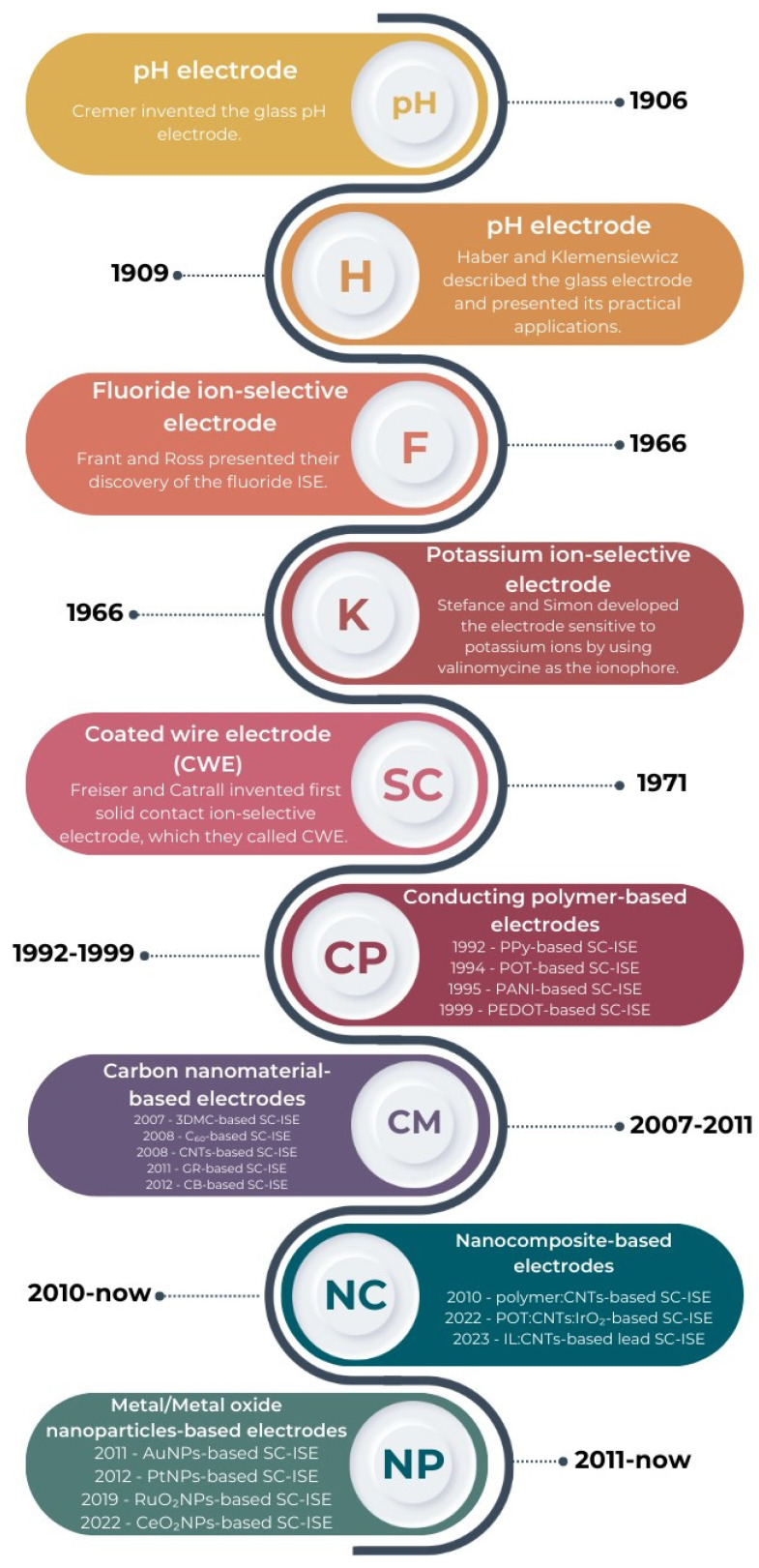
A brief history of ISEs development.

**Figure 2 materials-16-05779-f002:**
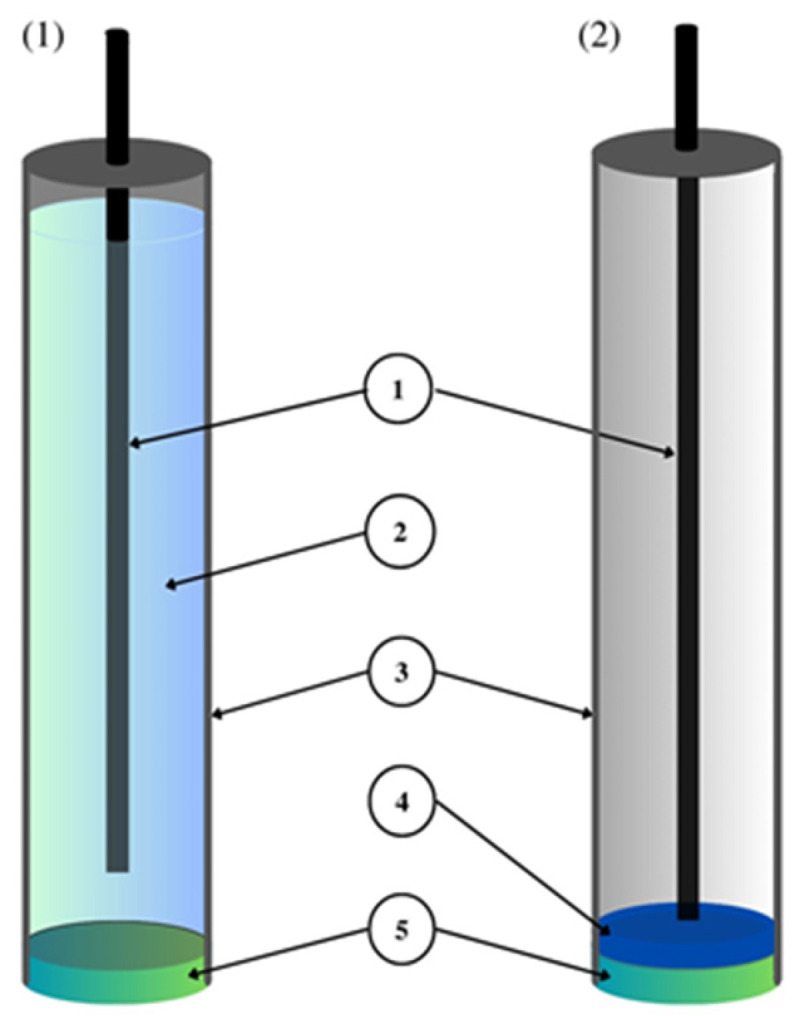
Construction of liquid contact ion-selective electrode (**1**) and solid contact ion-selective electrode (**2**) (1—inner electrode; 2—internal electrolyte; 3—holder; 4—solid contact layer; and 5—ion-selective membrane).

**Figure 3 materials-16-05779-f003:**
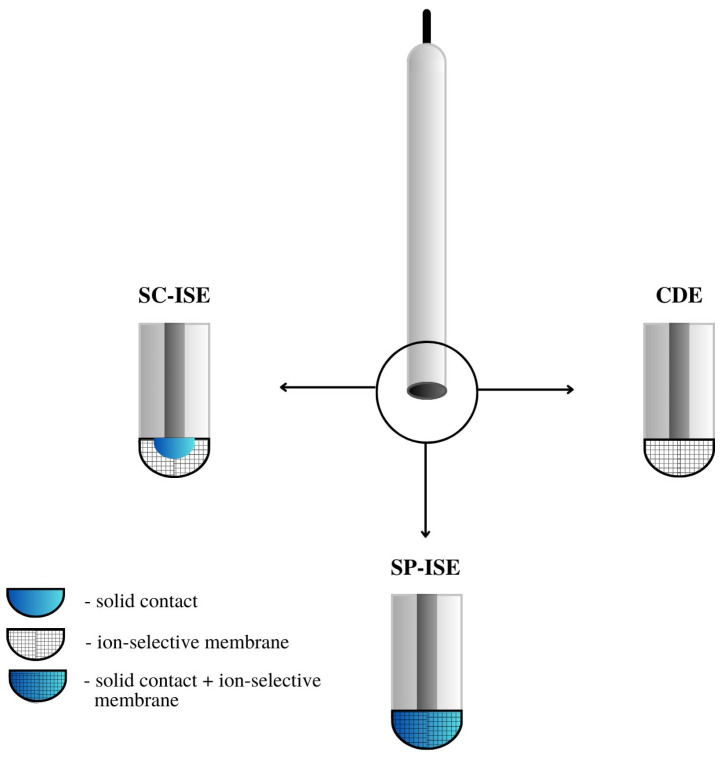
A comparison of the construction of CP-ISE, SP-ISE and SC-ISE electrodes.

**Figure 4 materials-16-05779-f004:**
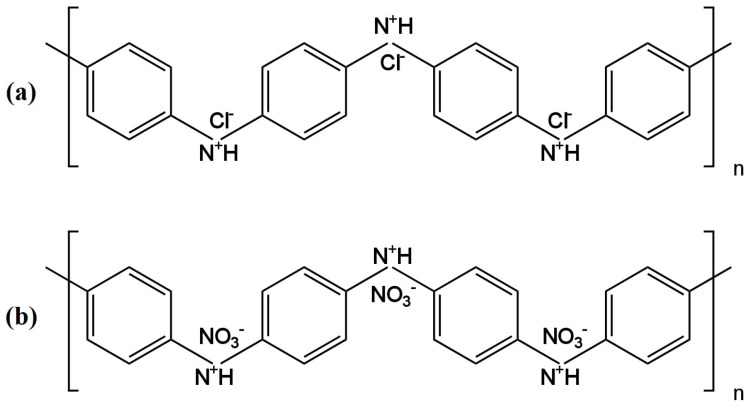
The structure of polyaniline doped with chloride (**a**) and nitrate (**b**) ions [26].

**Figure 5 materials-16-05779-f005:**
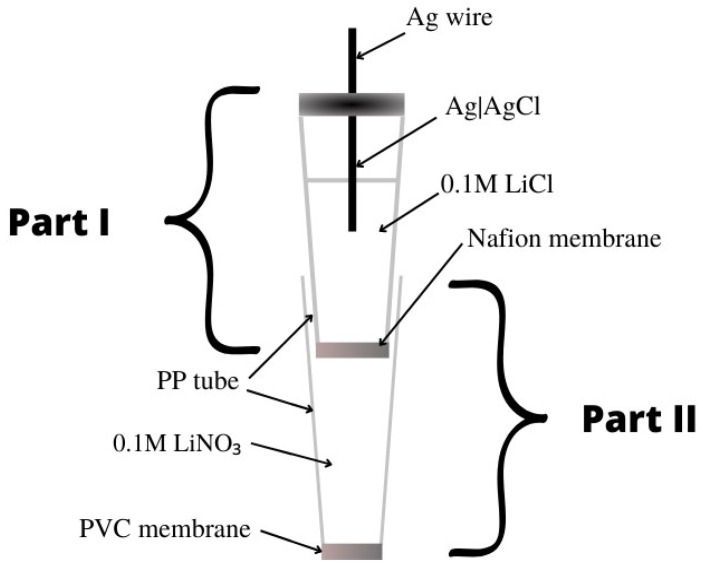
Ion-selective electrode construction based on [39].

**Figure 6 materials-16-05779-f006:**
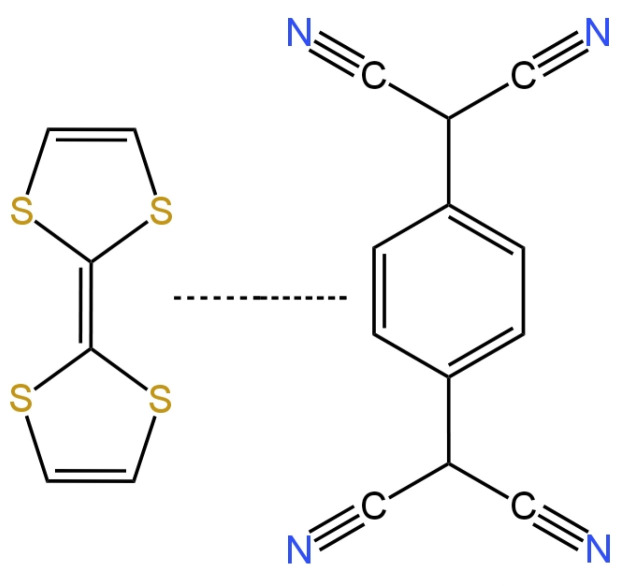
Tetrathiafulvalene 7,7,8,8-tetracyanoquinodimethane TTF-TCNQ salt used as solid contact in nitrate SC-ISE [40].

**Figure 7 materials-16-05779-f007:**
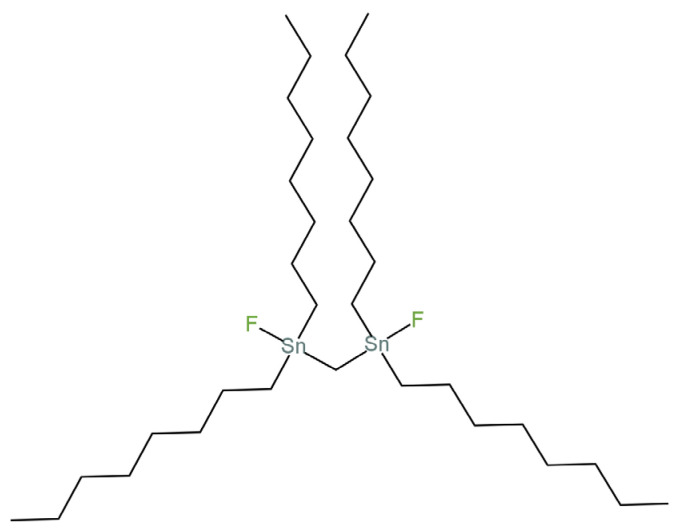
The structure of bis(fluorodioctylstannyl)methane.

**Figure 8 materials-16-05779-f008:**
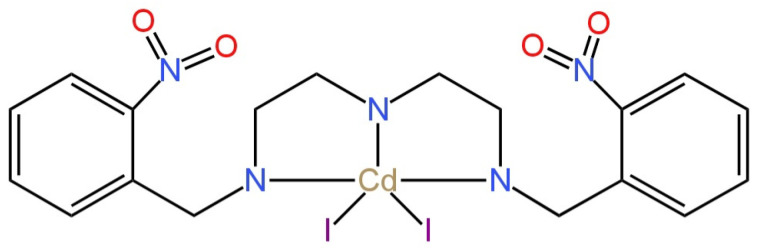
The structure of CdLI_2_—cadmium iodide complex [47].

**Figure 9 materials-16-05779-f009:**
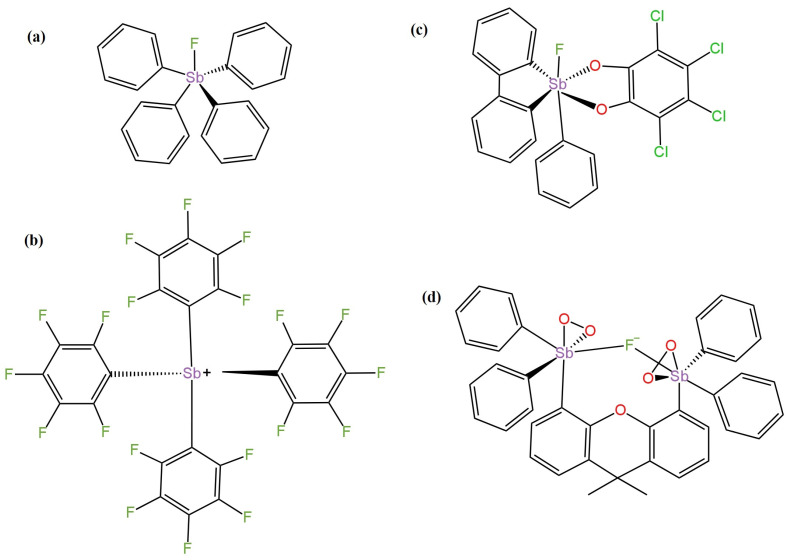
Lewis acidic organo-antimony(V) compounds used as fluoride ionophores: tetraphenyl stibonium fluoride (**a**), tetrakis-(pentafluorophenyl)stibonium (**b**), tetrachloro-substituted organoantimony(V) compound with fluoride (**c**) and bidentate organoantimony(V) compound with fluoride (**d**) [48].

**Figure 10 materials-16-05779-f010:**
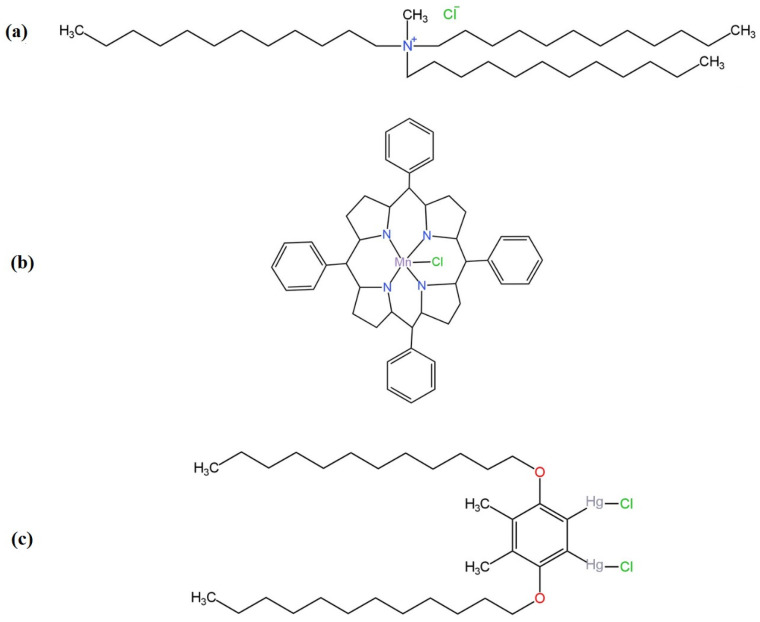
Active substances used in chloride ISEs: TDMACl (**a**), chloride ionophore(I) (**b**) and chloride ionophore(III) (**c**).

**Figure 11 materials-16-05779-f011:**
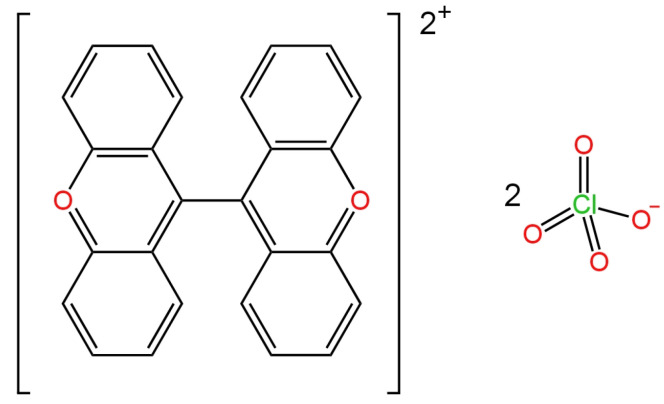
The structure of dixanthylium dye [59].

**Figure 12 materials-16-05779-f012:**
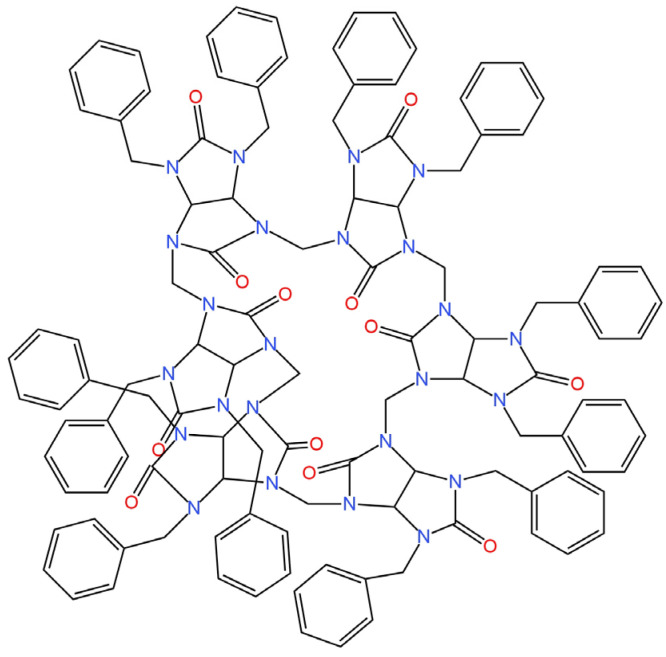
The structure of novel ionophore—Bn_12_BU [6].

**Figure 13 materials-16-05779-f013:**
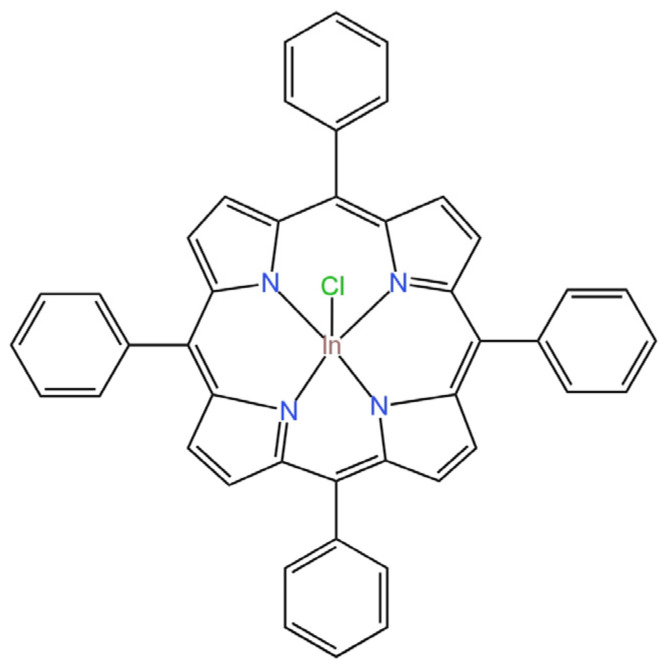
The structure of In^III^-porph.

**Figure 14 materials-16-05779-f014:**
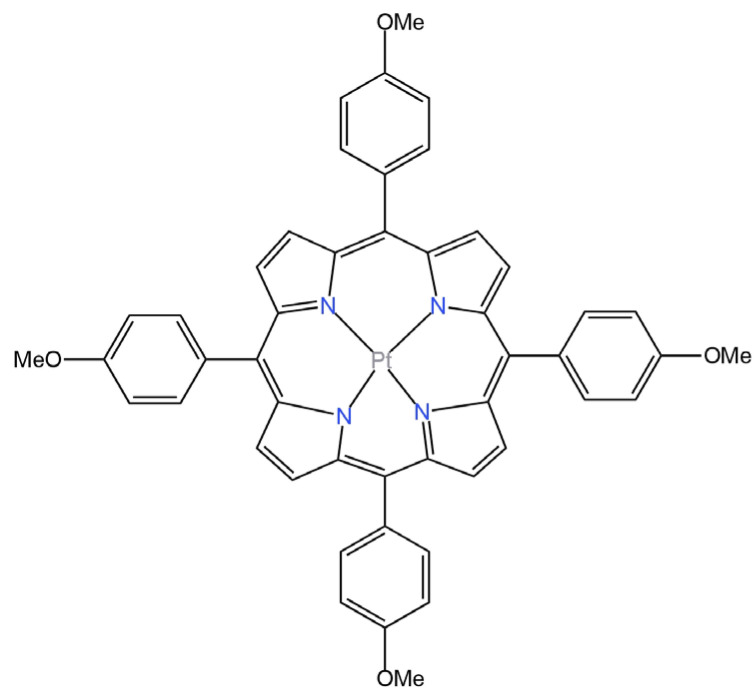
Structure of PtTMeOPP [65].

**Figure 15 materials-16-05779-f015:**
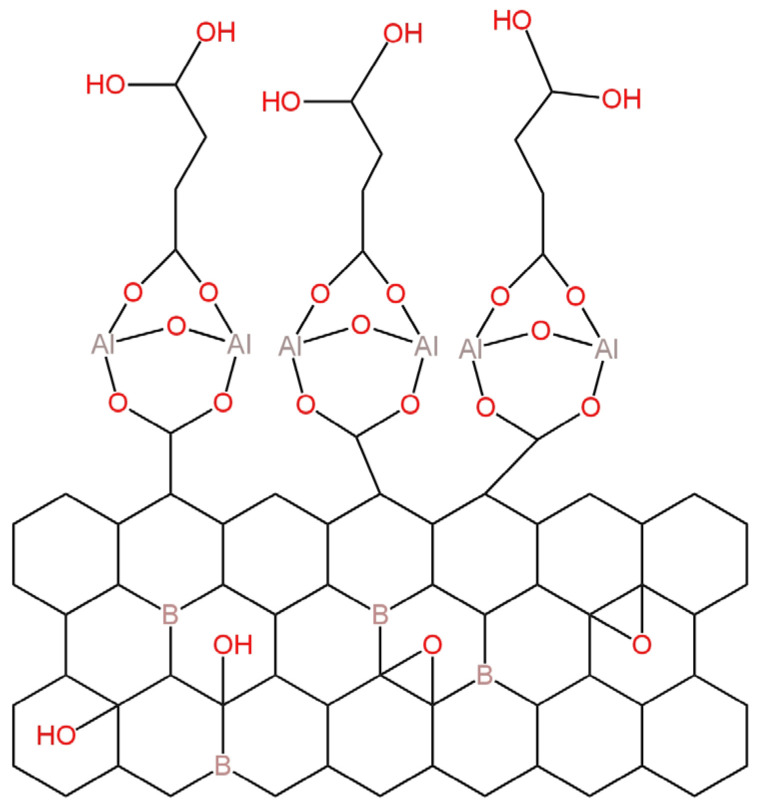
The structure of BGO/AlFu MOF [66].

**Figure 16 materials-16-05779-f016:**
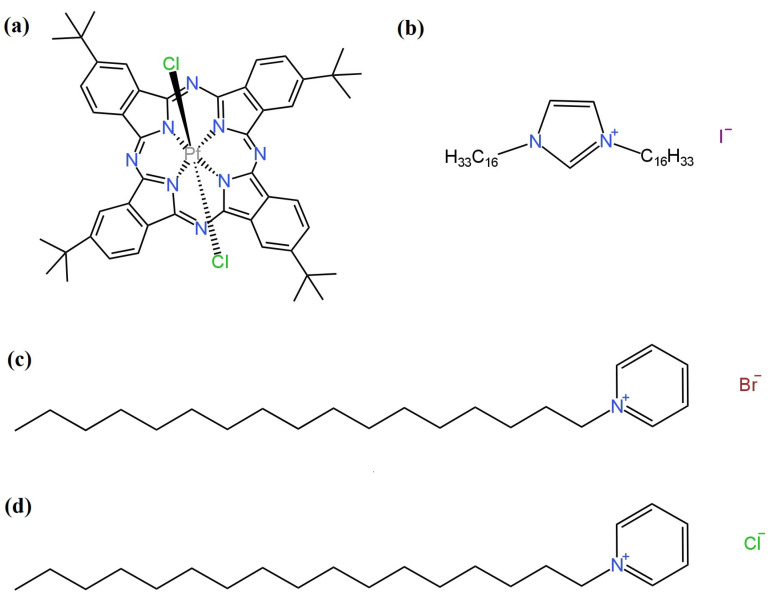
Structure of novel iodic ionophores: platinum (IV) tetra-tertbutylphthalocyanine dichloride (**a**), 1,3-dicetylimidazolium iodide (**b**), cetylpyridinuim bromide (**c**) and cetylpyridinuim chloride (**d**) [72].

**Figure 17 materials-16-05779-f017:**
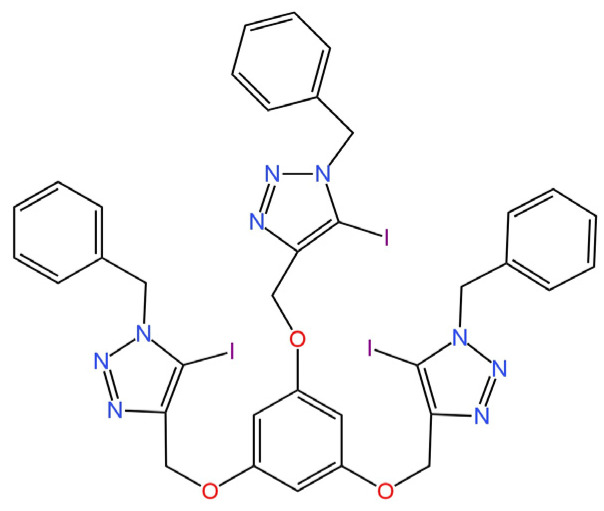
Structure of XB1 [73].

**Figure 18 materials-16-05779-f018:**
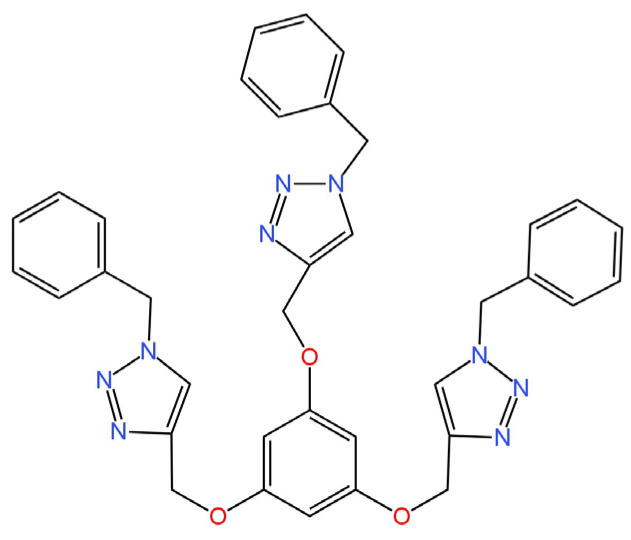
Structure of HB_1_ [73].

**Figure 19 materials-16-05779-f019:**
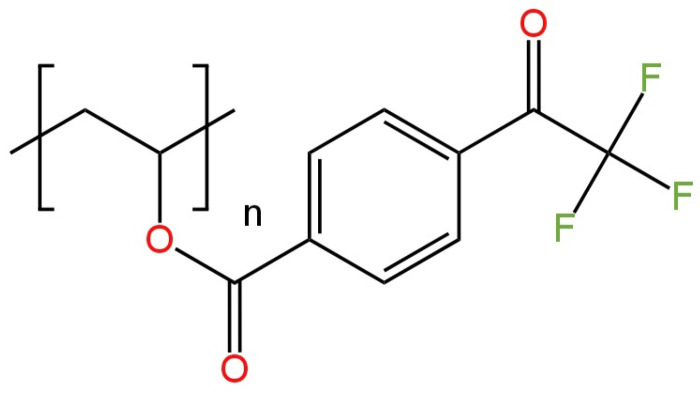
The structure of TFAB-PVC [77].

**Figure 20 materials-16-05779-f020:**
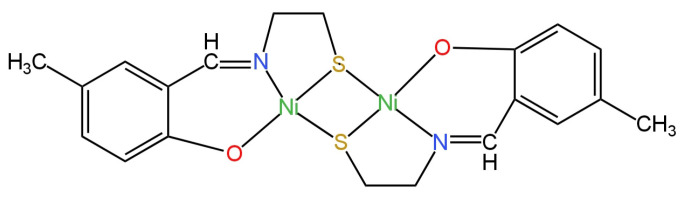
Schiff base complex with nickel used as a sulfate ionophore [80].

**Figure 21 materials-16-05779-f021:**
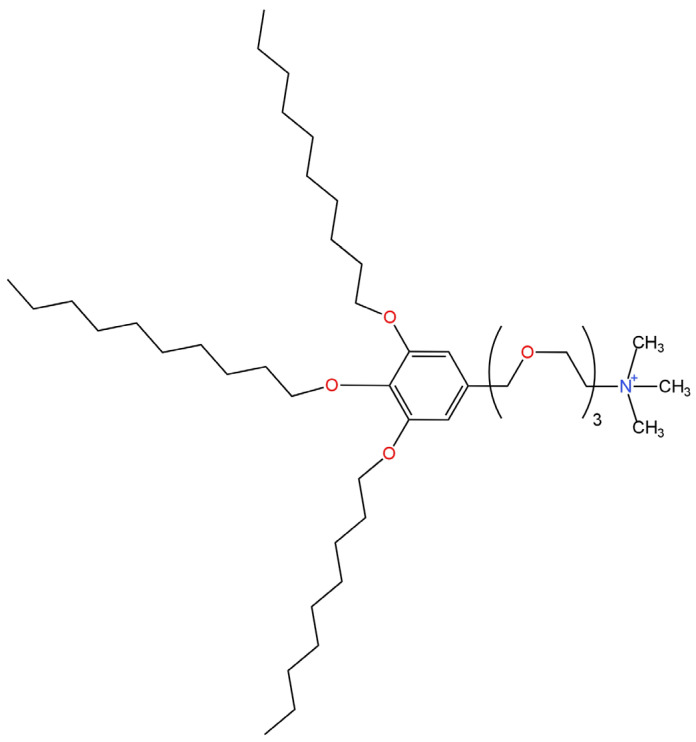
The structure of (oxyethyl)_3_TM.

**Figure 22 materials-16-05779-f022:**
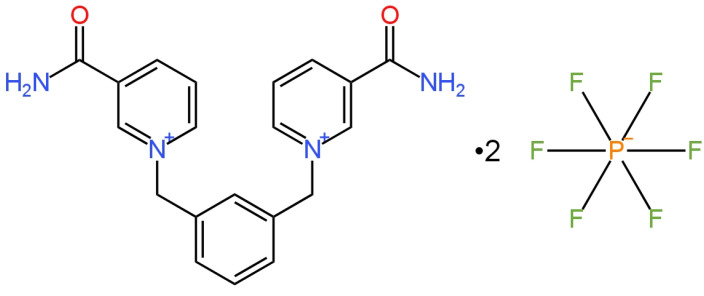
The structure of bis-meta-NICO-PF6 [86].

**Figure 23 materials-16-05779-f023:**
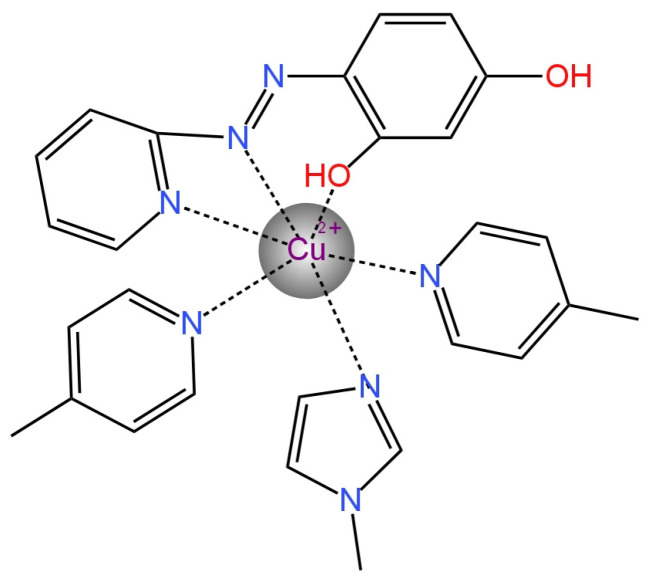
Cross-linked copper(II) doped copolymer [90].

**Figure 24 materials-16-05779-f024:**
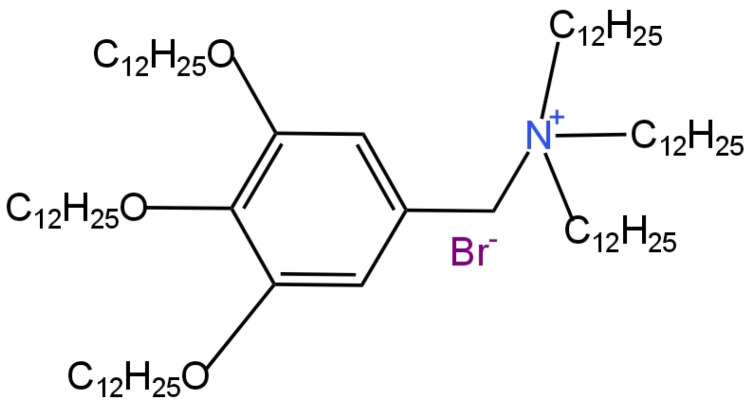
Structure of 3,4,5-tris(dodecyloxy)benzyltrilauryl ammonium (TL) bromide [99].

**Figure 25 materials-16-05779-f025:**
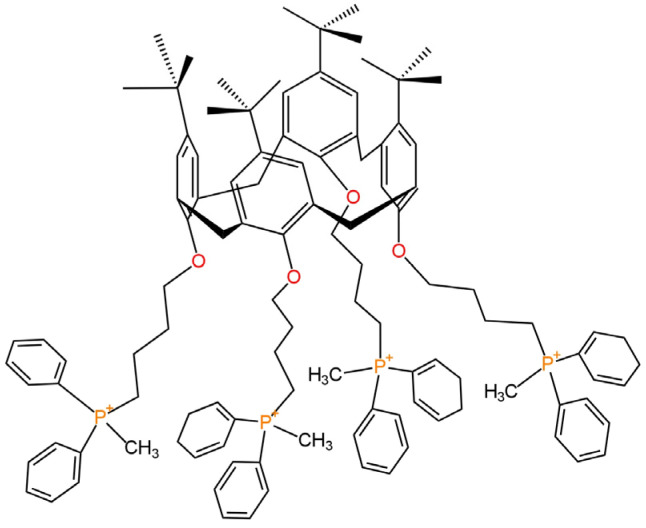
The structure of a new ionophore—tetrakis-(4-diphenylmethylphosphonium-butoxy)-tetrakis-p-tert-butylcalix [4]arene tetrathiocyanate [100].

**Figure 26 materials-16-05779-f026:**
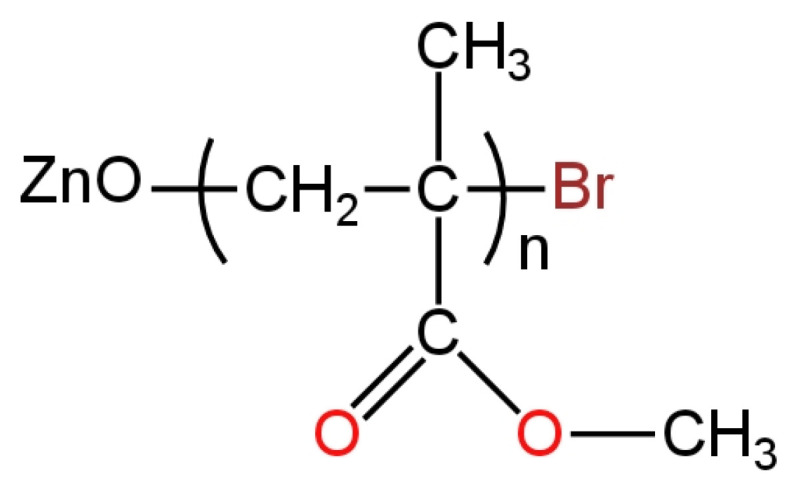
The structure of PMMA-ZnO [102].

**Figure 27 materials-16-05779-f027:**
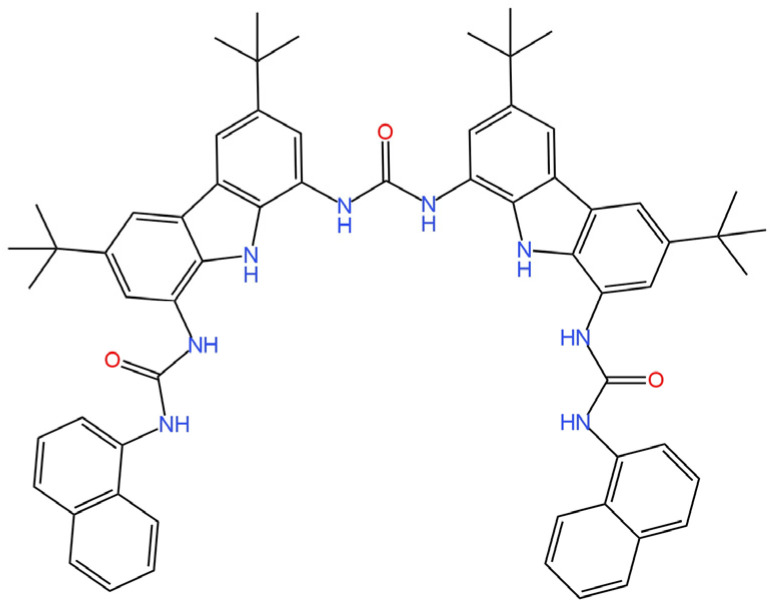
The structure of 1,3-bic(carbozyl)urea derivate [104].

**Table 1 materials-16-05779-t001:** Nitrate ion-selective electrodes.

E. No	Type of Contact	Ionophore/Ion Carrier	Intermediate/Transducer Layer	Type of Internal Electrode	Slope [mV/decade]	Range of Linearity [M]	Limit of Detection [M]	pH Range	Application/Samples	References
1	SC	TDMAN	Graphite	GrE	−55.4 ± 0.7	2.9 × 10^−4^–1.7 × 10^−1^	2.04 × 10^−4^	4.0–11.0	Industrial and environmental.	[23]
2	SC	TDMAN	LIG	-	−58.2 ± 4.2	5.0 × 10^−4^–1.0 × 10^−1^	6.01 × 10^−6^	6.0–8.0	Agricul-ture and surface water.	[24]
3	SC	TDMAN	Co_3_O_4_NPs	SPCE	−56.8	1.0 × 10^−7^–1.0 × 10^−2^	1.04 × 10^−8^	3.0–8.0	Aquaculture, river, domestic and tap water.	[28]
4	SC	TDMAN	PANINFs-Cl	GCE	−56.8	1.0 × 10^−6^–1.0 × 10^−1^	3.16 × 10^−7^	4.0–12.5	Environmental samples.	[26]
5	SC	TDMAN	PANINFs-NO_3_		−57.8	1.0 × 10^−6^–1.0 × 10^−1^	1.12 × 10^−6^	4.0–11.5		
6	SC	TDMAN	MCB	Ag wire	−54.8	5.0 × 10^−5^–1.0 × 10^−1^	2.5 × 10^−6^	-	-	[27]
7	SC	TDMAN	POT-MoS_2_	AuE	−64.0	7.1 × 10^−4^–1.0 × 10^−1^	9.2 × 10^−5^	-	Soil.	[25]
8	SC	PPy-NO_3_^−^	PGCP	GCE	−50.9	1.0 × 10^−5^–5.0 × 10^−1^	4.64 × 10^−6^	3.5–9.5	Environmental and clinical laboratories.	[29]
9	SC	PPy-NO_3_^−^	AuNPs	GCE	−50.4	5.3 × 10^−5^–1.0 × 10^−1^	5.25 × 10^−5^	-	Water samples and aqueous solutions of fertilizers.	[30]
10	SC	Nitrate ionophore VI	-	PAuE	−54.1	5.0 × 10^−5^–1.0 × 10^−1^	-	-	Field soils.	[31]
11	SC	Nitrate ionophore VI	TRGO	AuE	−60.0 ± 0.5	4.0 × 10^−5^–1.0 × 10^−1^	4.0 × 10^−6^	2.0–10.0	Blood.	[32]
12	SC	Nitrate ionophore VI	PTFE	SPCE	−58.0	-	-	4.0–11.0	Wastewater.	[33]
13	SC	Co(Bphen)_2_(NO_3_)_2_(H_2_O)_2_	MWCNTs-THTDPCl	GCE	−57.1	1.0 × 10^−6^–1.0 × 10^−1^	5.0 × 10^−7^	6.0–8.0	-	[35]
14	SC	Co(Bphen)_2_(NO_3_)_2_(H_2_O)_2_	Ag/AgCl/Cl^−^	Ag|AgCl	−56.3	1.0 × 10^−5^–1.0 × 10^−1^	3.98 × 10^−6^	5.4–10.6	Mineral, tap and river water.	[34]
15	SC	TDANO_3_	rGOA	SPCE	−59.1	1.0 × 10^−6^–1.0 × 10^−1^	7.59 × 10^−7^	-	Plant sap e.g., perilla leaf.	[36]
16	LC	TDANO_3_	0.01 M KNO_3_ and 0.001 M KCl	Ag|AgCl	−53.7 ± 0.4	1.0 × 10^−5^–1.0 × 10^−1^	1.3 × 10^−6^	-	-	[37]
17	SC	Nit^+^/NO_3_^−^	MWCNTs	GCE	−55.1 ± 1.0	8.0 × 10^−8^–1.0 × 10^−2^	2.8 × 10^−8^	3.5–10.0	Environmental samples.	[38]
18	LC	THANO_3_	0.1 M LiCl and 0.1 M LiNO_3_	Ag|AgCl	−53.3 ± 1.0	1.0 × 10^−5^–1.0 × 10^−1^	1.0 × 10^−6^	-	Hydroponic solutions.	[39]
19	SC	Nitrate ionophore V	TTF-TCNQ	GCD	−58.5	1.0 × 10^−5^–1.0 × 10^−1^	1.6 × 10^−6^	-	Aqueous samples.	[40]

**Table 2 materials-16-05779-t002:** Fluoride ion-selective electrodes.

E. No	Type of Contact	Ionophore/Ion Carrier	Intermediate/Transducer Layer	Type of Internal Electrode	Slope [mV/decade]	Range of Linearity [M]	Limit of Detection [M]	pH Range	Application/Samples	References
1	SC	Eu-doped LaF_3_ nanocrystals	-	TiE	−56 ± 13	1.0 × 10^−5^–1.0 × 10^−1^	1.0 × 10^−6^	-	-	[43]
2	SC	LaF_3_ single crystal	PEDOT	AgE	−56.0 ± 0.9	1.0 × 10^−5^–1.0 × 10^−1^	2.0 × 10^−5^	5.0–11.0	-	[44]
3	SC		Ag paste	AgE	−62.8 ± 3.8	1.0 × 10^−5^–1.0 × 10^−1^	1.0 × 10^−2^	-		
4	LC		PBS, 0.01 M Na_2_HPO_3_ and 0.02 M KH_2_PO_3_	Ag|AgCl	−38.6 ± 9.1	1.0 × 10^−5^–1.0 × 10^−1^	-	-		
5	SC	LaF_3_ single crystal	Fe_x_O_y_ NPs	SSDE	−52.9–−57.3	6.3 × 10^−6^–1.0 × 10^−1^	3.6 × 10^−7^	4.0–7.0	-	[45]
6	LC	LaF_3_ single crystal	KCl + HCl + 0.1 M AgNO_3_	Ag|AgCl	−50.8–−52.7	3.9 × 10^−7^–1.0 × 10^−1^	7.4 × 10^−8^	4.0–7.0		
7	SC	Bis(fluorodioctylstannyl)methane	MWCNTs-COOH	SPCE	−59.2	-	1.7 × 10^−9^	-	-	[46]
8	LC	tetrakis-(pentafluorophenyl)stibonium	0.2 M Gly/H_3_PO_4_ buffer and 0.001 M NaF	Ag|AgCl	−59.3	1.0 × 10^−5^–4.0 × 10^−2^	5.0 × 10^−6^	3.0	Tap water samples.	[48]
9	LC	[Ph_4_Sb]^+^			−58.2	1.0 × 10^−5^–4.0 × 10^−2^	2.0 × 10^−5^	-		
10	LC	tetrachloro-substituted organoantimony(V)			−54.6	1.0 × 10^−5^–4.0 × 10^−2^	2.0 × 10^−4^	-		
11	LC	Organo antimony(V) compound			−57.8	1.0 × 10^−5^–4.0 × 10^−2^	3.0 × 10^−5^	-		
12	SC	CdLI_2_	-	CPE	−58.9	1.5 × 10^−6^–5.5 × 10^−3^	1.2 × 10^−7^	5.0–7.0	River water samples.	[47]

**Table 3 materials-16-05779-t003:** Chloride ion-selective electrodes.

E. No	Type of Contact	Ionophore/Ion Carrier	Intermediate/Transducer Layer	Type of Internal Electrode	Slope [mV/decade]	Range of Linearity [M]	Limit of Detection [M]	pH Range	Application/Samples	References
1	SC	Chloride ionophore(III)	MWCNTs	GCE	−59.6	5.0 × 10^−6^–1.0 × 10^−1^	2.6 × 10^−6^	4.0–9.0	Inspection of the efficiency of water desalination.	[53]
2			PANINFs-Cl		−60.3		2.8 × 10^−6^			
3			PANINFs-MWCNTs		−61.2		2.7 × 10^−6^			
4	LC	TDMACl	0.5 M NaCl	Ag|AgCl	−55.0 ± 2	1.0 × 10^−5^–1.0 × 10^−1^	1.0 × 10^−6^	2.0–8.0	Wastewater samples.	[54]
5	SC	TDMACl	-	GCD	−57.1 ± 0.66	1.0 × 10^−4^–1.0 × 10^−1^	1.6 × 10^−5^	-	Water samples.	[55]
6			TTF		−58.2 ± 0.27	1.0 × 10^−5^–1.0 × 10^−1^	5.0 × 10^−6^			
7			TTF-TCNQ		−58.3 ± 0.16	1.0 × 10^−5^–1.0 × 10^−1^	4.0 × 10^−6^			
8			TTFCl		−58.4 ± 0.14	1.0 × 10^−5^–1.0 × 10^−1^	4.0 × 10^−6^			
9			CB		−59.6 ± 0.11	1.0 × 10^−5^–1.0 × 10^−1^	2.5 × 10^−6^			
10			CB-TTF		−58.5 ± 0.13	1.0 × 10^−5^–1.0 × 10^−1^	3.2 × 10^−6^			
11			CB-TTF-TCNQ		−58.7 ± 0.10	1.0 × 10^−5^–1.0 × 10^−1^	2.5 × 10^−6^			
12			CB-TTFCl		−59.1 ± 0.09	1.0 × 10^−5^–1.0 × 10^−1^	2.0 × 10^−6^			
13	SC	g-C_3_N_4_/AgCl	-	CPE	−55.4 ± 0.3	1.0 × 10^−6^–1.0 × 10^−1^	4.0 × 10^−7^	-	Aqueous samples.	[56]
14	SC with ISM	Chloride ionophore(I)	-	Ag|AgCl	−61.7 ± 2.4	1.0 × 10^−5^–1.0 × 10^−1^	1.1 × 10^−5^	-	Sweat.	[57]
15	SC	AgCl:Ag_2_S:PTFE	Fe_x_O_y_ NPs	multi-purpose solid state electrode made from stainless steel	−44.4	2.0 × 10^−6^–1.0 × 10^−1^	1.42 × 10^−6^	-	-	[58]
16			ZnO NPs		−40.5	3.6 × 10^−6^–1.0 × 10^−1^	1.0 × 10^−6^	-		

**Table 4 materials-16-05779-t004:** Perchlorate ion-selective electrodes.

E. No	Type of Contact	Ionophore/Ion Carrier	Intermediate/Transducer Layer	Type of Internal Electrode	Slope [mV/decade]	Range of Linearity [M]	Limit of Detection [M]	pH Range	Application/Samples	References
1	SC	Dixanthylium dye	-	Pt wire	−57.4	1.0 × 10^−6^–6.1 × 10^−2^	5.0 × 10^−7^	1.5–11.0	Aqueous samples.	[59]
2	SC	Bn_12_BU [6]	PEDOT	GCE	−59.9 ± 1.1	1.0 × 10^−6^–1.0 × 10^−1^	1.0 × 10^−6^	-	Real water samples.	[60]
3	SC	In^III^-porphyrin	SWCNTs	GCE	−56.0 ± 1.1	1.1 × 10^−6^–1.0 × 10^−2^	1.8 × 10^−7^	-	Fireworks and propellants.	[61]

**Table 5 materials-16-05779-t005:** Bromide ion-selective electrodes.

E. No	Type of Contact	Ionophore/Ion Carrier	Intermediate/Transducer Layer	Type of Internal Electrode	Slope [mV/decade]	Range of Linearity [M]	Limit of Detection [M]	pH Range	Application/Samples	References
1	LC	TDMABr	0.5 M NaBr	Ag|AgCl	−57.4 ± 0.3	1.0 × 10^−6^–1.0 × 10^−1^	1.2 × 10^−6^	1.0–11.0	Water samples e.g., tea samples.	[54]
2	LC	BGO/AlFu-MOF	0.05 M KBr	GE	−54.5 ± 0.2	1.0 × 10^−7^–1.0 × 10^−1^	7.1 × 10^−8^	4.0–9.0	Environmental samples.	[66]
3	SC	Mesotetraphenylporphyrin manganese(III)-chloride complex	POT	GCE	-	1.0 × 10^−6^–1.0 × 10^−2^	2.0 × 10^−9^	-	Water samples.	[67]
4		4,5-dimethyl-3,6-dioctyloxy-o-phenylene-bis(mercurytrifluoroacetate)								
5	LC	PtTMeOPP	0.01 M KCl	Ag|AgCl	−64.4 ± 0.4	1.0 × 10^−5^–1.0 × 10^−1^	8.0 × 10^−6^	6.0–12.0	Pharmaceutical samples.	[65]

**Table 6 materials-16-05779-t006:** Iodide ion-selective electrodes.

E. No	Type of Contact	Ionophore/Ion Carrier	Intermediate/Transducer Layer	Type of Internal Electrode	Slope [mV/decade]	Range of Linearity [M]	Limit of Detection [M]	pH Range	Application/Samples	References
1	LC	TDMAI	0.1 M KI and 0.1 M KCl	Ag|AgCl	−54 ± 1	1.0 × 10^−5^–1.0 × 10^−1^	1.3 × 10^−6^	2.0–8.0	Wastewater samples.	[54]
2	SC	XB1	PANI	SPE	−54 ± 1	1.0 × 10^−5^–1.0 × 10^−1^	1.3 × 10^−6^	-	-	[73]
3		HB2			−51.9	1.0 × 10^−6^–1.0 × 10^−1^	1.3 × 10^−6^			
4	LC	Pc^t^PtCl_2_	0.001M KI and 0.1 KCl	Ag|AgCl	−54.9	1.0 × 10^−6^–1.0 × 10^−1^	1.0 × 10^−6^	-	Medicaments such as “Iodomarine 100” and other pharmaceuticals.	[72]
5		CPCl + Pc^t^PtCl_2_			−26 ± 3	1.0 × 10^−3^–1.0 × 10^−1^	1.8 × 10^−4^			
6		CPBr + Pc^t^PtCl_2_			−45 ± 1	1.0 × 10^−4^–1.0 × 10^−1^	2.1 × 10^−5^			
7		CPBr			−54 ± 1	1.0 × 10^−4^–1.0 × 10^−1^	3.5 × 10^−5^			
8	SC	CPCl + Pc^t^PtCl_2_	graphite	SPPE	−46 ± 2	1.0 × 10^−3^–1.0 × 10^−1^	3.0 × 10^−4^			
9		CPBr + Pc^t^PtCl_2_			−51 ± 1	1.0 × 10^−4^–1.0 × 10^−1^	5.3 × 10^−5^			
10		CPBr			−54 ± 1	1.0 × 10^−4^–1.0 × 10^−1^	1.9 × 10^−5^			
11		DCImI + Pc^t^PtCl_2_			−50 ± 1	1.0 × 10^−3^–1.0 × 10^−1^	1.0 × 10^−4^			
12	LC	AgCl:Ag_2_S:PTFE + ZnO NPs	-	-	−57 ± 2	1.0 × 10^−4^–1.0 × 10^−1^	1.8 × 10^−5^	-	Penicillamine in real samples.	[71]
13	LC	PtTMeOPP	0.01M KCl	Ag|AgCl	−57.4 ± 0.3	2.5 × 10^−6^–1.0 × 10^−2^	2.2 × 10^−6^	3.0–12.0	Pharmaceutical such as potassium iodide tablets.	[65]

**Table 7 materials-16-05779-t007:** Ion-selective electrodes sensitive to S^2−^, SO_3_^2−^ and SO_4_^2−^ ions.

E. No	Ion	Type of Contact	Ionophore/Ion Carrier	Intermediate/Transducer Layer	Type of Internal Electrode	Slope [mV/decade]	Range of Linearity [M]	Limit of Detection [M]	pH Range	Application/Samples	References
1	S^2−^	LC	Ag_2_S	10^−6^ M Na_2_S	Ag|AgCl	−28.2	1.0 × 10^−6^–1.0 × 10^−1^	2.3 × 10^−7^	6.0–12.0	Industrial water, e.g., petroleum industries.	[76]
2	S^2−^	SC	Ag_2_S	RGSs	Ag wire	-	5.0 × 10^−7^–1.0 × 10^−3^	1.8 × 10^−7^	-	Sea and tap water.	[78]
3	S^2−^	LC	TFAB-PVC	0.01 M Na2S and 0.001 M KCl	-	−27.1	1.0 × 10^−6^–1.0 × 10^−1^	6.0 × 10^−7^	-	Water samples.	[77]
4			TFABAHE			−27.1	1.0 × 10^−6^–1.0 × 10^−1^	3.8 × 10^−7^			
5	SO_3_^2−^	SC	CoPC	MWCNTs-COOH	SPCE	−29.8 ± 0.4	2.0 × 10^−6^–2.3 × 10^−3^	1.1 × 10^−6^	5.0–7.2	Various samples.	[79]
6	SO_3_^2−^	SC		PANINFs		−26.5 ± 0.6	5.0 × 10^−6^–2.3 × 10^−3^	1.5 × 10^−6^	4.8–7.7		
7	SO_4_^2−^	SC	Shiff base complex with nickel	-	CPE	−29.7	7.5 × 10^−9^–1.5 × 10^−3^	5.0 × 10^−9^	3.0–9.0	Water and blood sam-ples.	[80]
8	SO_4_^2−^	LC	(oxyethyl)_3_TM	0.01 M Na_2_SO_4_ and 0.001 M KCl	-	−27.0	^−^	6.7 × 10^−7^	-	Water samples.	[81]
9	SO_4_^2−^	LC	TFAB-PVC	0.01 M Na_2_S and 0.001 M KCl	-	−25.7	1.0 × 10^−6^–1.0 × 10^−2^	1.0 × 10^−6^	-	Water samples.	[77]
10	SO_4_^2−^		TFABAHE		-	−26.5	1.0 × 10^−6^–1.0 × 10^−2^	7.0 × 10^−7^	-		

**Table 8 materials-16-05779-t008:** Ion-selective electrodes sensitive to phosphate ions.

E. No	Ion	Type of Contact	Ionophore/Ion Carrier	Intermediate/Transducer Layer	Type of Internal Electrode	Slope [mV/decade]	Range of Linearity [M]	Limit of Detection [M]	pH Range	Application/Samples	References
1	H_2_PO_4_^−^	LC	*Bis-meta*-NICO-PF_6_	-	-	−53.3	1.0 × 10^−6^–1.0 × 10^−2^	0.9 × 10^−6^	-	Environmental and other real samples.	[86]
2	H_2_PO_4_^−^	SC	Co-PPY-OMC	-	GCE	−31.6	1.0 × 10^−5^–5.0 × 10^−2^	6.8 × 10^−6^	3.0–5.0	Water samples for example in human urine or wastewater.	[87]
3	H_2_PO_4_^−^	SC	nano-IIP	-	CPE	−30.6 ± 0.5	1.0 × 10^−5^–1.0 × 10^−1^	4.0 × 10^−6^	9.0–12.0	Water samples.	[88]
4	HPO_4_^2−^	SC	Ba_3_PO_4_ + Cu_2_S + Ag_2_S pellet	-	Cu wire	−57.0	1.0 × 10^−6^–1.0 × 10^−1^	2.4 × 10^−7^	7.0–9.0	Food samples e.g., meat, vegetables and fruits.	[89]
5	HPO_4_^2−^	SC	Cu(II)-DCP	MWCNTs + graphite	Cu wire	−30.7 ± 0.4	1.0 × 10^−6^–1.0 × 10^−1^	6.5 × 10^−7^	-	Water samples.	[90]
6	HPO_4_^2−^	SC	BiPO_4_	Bi particles	Pt wire	−30.3	1.0 × 10^−6^–1.0 × 10^−1^	7.7 × 10^−7^	5.0–9.0	Drinking water.	[91]
7	HPO_4_^2−^	SC	MoO_2_ + PMo_12_O_40_^3−^	-	Mo wire	−27.8 ± 0.5	1.0 × 10^−5^–1.0 × 10^−1^	1.0 × 10^−6^	8.0–9.5	Wastewater, nutrient solution and Coca-Cola.	[92]
8	HPO_4_^2−^	SC	Ag_3_PO_4_ + Ag_2_S	PTFE	SSD	−21.0	1.0 × 10^−5^–1.0 × 10^−1^	5.3 × 10^−6^	3.0–7.0	Solution of pH range 3–7.	[93]
9				MWCNTs	Cu wire	−32.6	1.0 × 10^−5^–1.0 × 10^−1^	5.5 × 10^−6^			
10	HPO_4_^2−^		TFAB-PVC	0.01 M Na_2_S and 0.001 M KCl	-	−27.5	1.0 × 10^−7^–1.0 × 10^−2^	7.0 × 10^−7^	-	Water samples	[77]
11			TFABAHE			−28.7	1.0 × 10^−7^–1.0 × 10^−2^	5.0 × 10^−7^			
12	PO_4_^3−^	LC	IIP-1 (chitosan-La(III)-PO_4_^3−^)	0.001 M KCl + 0.001 M Na_3_PO_4_	Cu wire	−3.2	1.0 × 10^−6^–1.0 × 10^−2^	7.6 × 10^−6^	5.0–7.0	Household wastewater.	[94]
13	PO_4_^3−^	LC	IIP-2 (chitosan-La(III)-AAPTS-PO_4_^3−^)			−1.9	1.0 × 10^−6^–1.0 × 10^−2^	5.1 × 10^−6^			
14	PO_4_^3−^	LC	IIP-3 (AAPTS-La(III)-PO_4_^3−^)			−3.7	1.0 × 10^−6^–1.0 × 10^−2^	2.5 × 10^−6^			

**Table 9 materials-16-05779-t009:** Ion-selective electrodes sensitive to SCN^−^ ions.

E. No	Type of Contact	Ionophore/Ion Carrier	Intermediate/Transducer Layer	Type of Internal Electrode	Slope [mV/decade]	Range of Linearity [M]	Limit of Detection [M]	pH Range	Application/Samples	References
1	LC	TL	0.01 M KSCN	-	−53.9	1.0 × 10^−6^–1.0 × 10^−1^	5.6 × 10^−6^	0.5–12.5	Human saliva.	[99]
2	LC	Tetrakis-(4-diphenylmethylphosphonium-butoxy)-tetrakis-p-tert-butylcalix [4]arene tetrathiocyanate	0.001 M KCl	Ag|AgCl	−55.5 ± 2.1	1.0 × 10^−5^–1.0 × 10^−1^	6.3 × 10^−6^	-	Saliva and other medical measurements.	[100]
3	SC		-	GCE	−59.9 ± 0.3	1.0 × 10^−6^–1.0 × 10^−1^	1.6 × 10^−6^			
4	SC		-	Au rods	−53.3 ± 0.3	1.0 × 10^−6^–1.0 × 10^−1^	3.2 × 10^−6^			
5	LC	Aliquat336-SCN	0.01 M NaSCN and 0.1 M NaCl	Ag|AgCl	−56.3	3.2 × 10^−5^–5.0 × 10^−1^	6.3 × 10^−6^	-	Human saliva.	[101]

**Table 10 materials-16-05779-t010:** Ion-selective electrodes sensitive to other ions.

E. No	Ion	Type of Contact	Ionophore/Ion Carrier	Intermediate/Transducer Layer	Type of Internal Electrode	Slope [mV/decade]	Range of Linearity [M]	Limit of Detection [M]	pH Range	Application/Samples	References
1	AsO_4_^3−^	LC	PMMA-ZnO	0.05 M Na_3_AsO_4_	SCE	−28.6	1.0 × 10^−9^–1.0 × 10^−1^	1.0 × 10^−9^	4.0–7.0	Water solutions.	[102]
2	BO_3_^3−^	SC	Ag_2_B_4_O_7_	MWCNTs	Cu wire	−34.0 ± 1.0	1.0 × 10^−4^–1.0 × 10^−1^	5.6 × 10^−5^	5.0–8.0	Rock, soil.	[103]
3	CH_3_COO^−^	SC	1,3-bis(carbazolyl)urea	PEDOT	GCE	−51.3	3.2 × 10^−5^–7.9 × 10^−2^	1 × 10^−5^	6.0–8.0	Aqueous samples.	[104]
4	CO_3_^2−^	SC	Carbonate ionophore VII	Carbon film	Ni wire	−30.4	1.0 × 10^−5^–1.0 × 10^−1^	2.8 × 10^−6^	-	Exploration of deep-sea hydrothermal activity.	[105]
5	SiO_3_^2−^	SC	PbSiO_3_	PbSiO_3_	Ag wire coated by the Pb film	−31.3	1.0 × 10^−5^–1.0 × 10^−1^	-	-	Aqueous samples with low-chloride content.	[106]

## Data Availability

Not applicable.

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
