# Peer review of "New Materials Used for the Development of Anion-Selective Electrodes—A Review"

_materials, 2023, doi:10.3390/ma16175779_

Round 1

Reviewer 1 Report

The author describes the “New materials used for the development of anion-selective electrodes - a review”. This paper is quite interesting from a technological point of view. However, there are still some key issues that need to be addressed. The author should revise their manuscript based on the comments and suggestions. I recommended a Major revision of the manuscript.

1.      Poor introduction writing. The author should address the present trend of anion-selective electrodes, and their advantages, and criticize previous reports by stating what needs to be done instead to improve the performance.

2.      The author mentioned “Ion selective electrodes are a popular analytical tool useful in the analysis of cations and anions”, however, there is no detail available for the cation.

3.      How will this work be helpful for revolutionizing the new class of materials for anion-selective electrodes?

4.      What is the advantage of the SC-ISEs over classical ISEs, if both have similar ion-selective membranes and inner electrode?

5.      How do the nitrates ions affect the perforce of Ion-selective electrode?

6.      What is their response to the carbides, iodide, sulfide, cyanide, chloride, or phosphides?

7.      Among AglAgCl, HglHgO, and SCE, which one is better for ion selective electrodes? Why the author choose AglAgCl?

8.      What are the main characteristics of the material for ion-selective electrode use?

9.      How does the ionophore play a role in varying the efficiency of ion-selective electrode?

10.  The author please check the grammatical/typo errors in the manuscript.

The author please check the grammatical/typo errors in the manuscript.

Author Response

Thank you very much for the valuable comments

The author describes the “New materials used for the development of anion-selective electrodes - a review”. This paper is quite interesting from a technological point of view. However, there are still some key issues that need to be addressed. The author should revise their manuscript based on the comments and suggestions. I recommended a Major revision of the manuscript.

  1. Poor introduction writing. The author should address the present trend of anion-selective electrodes, and their advantages, and criticize previous reports by stating what needs to be done instead to improve the performance.

 In line with the Reviewer’s comment, the introduction was improved and extended.

The present research trend in the area of anion-selective electrodes were presented in the introduction:

“The research in the area of ion-selective electrodes include two main research directions: the development of new active substances to obtain more selective ion-sensitive membranes and/or new, primarily electroconductive materials, which are used in the construction of electrodes without internal electrolyte solution to improve the process of charge transfer between the membrane and the internal electron conductor.   the present trend of anion-selective electrodes”.

We also discussed a recently published review article and pointed out the merits of our work.  

  1. The author mentioned “Ion selective electrodes are a popular analytical tool useful in the analysis of cations and anions”, however, there is no detail available for the cation.

There is no detail available for the cations, because in this paper we focused on the anion-selective electrodes. However, some general information contained in the work, e.g. a comparison of the construction of electrodes with liquid contact and with solid contact, as well as types of all solid state constructions, apply to both cation and anion-selective electrodes.

  1. How will this work be helpful for revolutionizing the new class of materials for anion-selective electrodes?

The aim of our review was to present the current state of knowledge in the field of ion-selective electrodes sensitive to inorganic anions. It is a comprehensive overview containing the characteristics of new materials used in the construction of ISEs, including both materials used as active substances of the ion-selective membrane as well as materials used to improve the efficiency and/or easier and more universal use of ISEs.

  1. What is the advantage of the SC-ISEs over classical ISEs, if both have similar ion-selective membranes and inner electrode?

In line with the Reviewer’s comment, the advantages of the SC-ISEs over classical ISEs were described in the introduction.

  1. How do the nitrates ions affect the perforce of Ion-selective electrode?
  2. What is their response to the carbides, iodide, sulfide, cyanide, chloride, or phosphides?

Response to the comments 5 and 6

The influence of a given ion on the performance of the electrode depends on the type of membrane and its active ingredient (ionophore), which gives it the sensitivity and selectivity for a specific ion. If the ionophore present in the membrane is characterized by high selectivity, then interfering ions have little effect on the electrode response. In the case of using an ion-exchanger as an active component of the membrane, its selectivity depends on the nature of the ion (more precisely, on its hydration energy).  

  1. Among AglAgCl, HglHgO, and SCE, which one is better for ion selective electrodes? Why the author choose AglAgCl?

All electrodes listed here are reference electrodes. Ag/AgCl is the most frequently chosen electrode for measurements with ISEs due to its ease of use and safety (absence of toxic mercury).

  1. What are the main characteristics of the material for ion-selective electrode use?

It depends on the function that the material is supposed to perform in the ion-selective electrode.

Materials used as ionophores should have high affinity for one ion and at the same time weakly interact with other ions. In this case, the electrode will show good selectivity towards the given ion, for which the affinity of the ionophore is high. For materials used as solid contact, high capacitance and low resistance are desirable. In any case, the material should be hydrophobic. These information are given in the corrected manuscript in chapter 2.

  1. How does the ionophore play a role in varying the efficiency of ion-selective electrode?

In response to the Reviewer's question at the beginning of chapter 2, additional clarification regarding the role of the ionophore was added.

  1. The author please check the grammatical/typo errors in the manuscript.

In line with the Reviewer’s comment, the entire article was re-checked and previously unnoticed errors were corrected.

 We hope our manuscript is now suitable for publication in Materials.

Reviewer 2 Report

In this manuscript, the authors overviewed new materials used for the preparation of anion-sensitive ion-selective electrodes in five recent years. The manuscript is full of content and well written. Thus, I suggest its publication after a minor revision. Some comments are below:

1.        It is unnecessary to provide two sets of identical pictures in Figure 1.

2.        It is suggested to provide a picture of brief history of ISEs for better comprehension.

3.        In the introduction section, more description is needed to present the advantages and disadvantages of classic ISEs and SC-ISEs, which would have facilitated the reader's understanding of the future research directions.

4.        Can you share some points on the development prospects and challenges for anion-sensitive ion-selective electrodes.

Author Response

Thank you very much for the valuable comments

In this manuscript, the authors overviewed new materials used for the preparation of anion-sensitive ion-selective electrodes in five recent years. The manuscript is full of content and well written. Thus, I suggest its publication after a minor revision. Some comments are below:

  1. It is unnecessary to provide two sets of identical pictures in Figure 1.

The Reviewer’s comment is absolutely right. Figure 1 ( in the revised manuscript it is Figure 2 ) has been corrected.

  1. It is suggested to provide a picture of brief history of ISEs for better comprehension.

In line with the Reviewer’s comment the brief history of ISEs is presented on the new Figure 1.

  1. In the introduction section, more description is needed to present the advantages and disadvantages of classic ISEs and SC-ISEs, which would have facilitated the reader's understanding of the future research directions.

In line with the Reviewer’s comment the introduction was extended. The advantages and disadvantages of classic ISEs and SC-ISEs were described more detailed.

  1. Can you share some points on the development prospects and challenges for anion-sensitive ion-selective electrodes.

In line with the Reviewer’s comment the development prospects and challenges for anion-sensitive ion-selective electrodes were discussed in the Conclusions.

Round 2

Reviewer 1 Report

The ms can be accepted in its current form.